# 2 and 3-dimensional structure of the descent of mesospheric trace constituents after the 2013 SSW elevated stratopause event

David E. Siskind[1], V. Lynn Harvey[2], Fabrizio Sassi[1], John P. McCormack[1,6], Cora E. Randall[2,3], Mark E. Hervig[4], and Scott. M. Bailey[5]

[1]Space Science Division, Naval Research Laboratory, Washington DC, USA
[2]Laboratory for Atmospheric and Space Physics, University of Colorado, Boulder CO, USA
[3]Department of Atmospheric and Oceanic Sciences, University of Colorado, Boulder CO, USA
[4]GATS Inc., Driggs, ID, USA
[5]Bradley Department of Electrical and Computer Engineering, Virginia Tech, Blacksburg VA, USA
[6]Now at Heliophysics Division, National Aerononautics and Space Administration, Washington DC, USA

**Correspondence:** David Siskind (david.siskind@nrl.navy.mil)

**Abstract.** We use the Specified Dynamics version of the Whole Atmosphere Community Climate Model Extended (SD-WACCMX) to model the descent of nitric oxide (NO) and other mesospheric tracers in the extended, elevated stratopause phase of the 2013 Sudden Stratospheric Warming (SSW). The dynamics are specified with a high altitude version of the Navy Global Environmental model (NAVGEM-HA). Consistent with our earlier published results, we find that using a high altitude
meteorological analysis to nudge WACCMX allows for a realistic simulation of the descent of lower thermospheric nitric oxide down to the lower mesosphere, near 60 km. This is important because these simulations only included auroral electrons, and did not consider additional sources of NO from higher energy particles that might directly produce ionization, and hence nitric oxide, below 80-85 km. This suggests that the so-called energetic particle precipitation indirect effect (EPP-IE) can be accurately simulated, at least in years of low geomagnetic activity, such as 2013, without the need for additional NO production,
provided the meteorology is accurately constrained. Despite the general success of WACCMX in bringing upper mesospheric NO down to 55-60 km, a detailed comparison of the WACCMX fields with the analyzed NAVGEM-HA $H_2O$ and satellite NO and $H_2O$ data from the Solar Occultation for Ice Experiment (SOFIE) and the Atmospheric Chemistry Experiment-Fourier Transform Spectrometer (ACE-FTS) reveals significant differences in the latitudinal and longitudinal distributions at lower altitudes. This stems from the tendency for WACCMX descent to maximize at sub-polar latitudes; and while such sub-polar
descent is seen in the NAVGEM-HA analysis, it is more transient than in the WACCMX simulation. These differences are linked to differences in the Transformed Eulerian Mean (TEM) circulation between NAVGEM-HA and WACCMX, most likely arising from differences in how gravity wave forcing is represented. To attempt to compensate for the differing distributions of model vs. observed NO and to enable us to quantify the total amount of upper atmospheric NO delivered to the stratopause region, we use potential vorticity and equivalent latitude coordinates. Preliminary results suggest both model and observations
are generally consistent with NO totals in the range of 0.1-0.25 gigamoles (GM).

# 1 Introduction

One of the more interesting and challenging problems in middle atmospheric science has been to accurately represent the descent of upper mesospheric and thermospheric nitric oxide ($NO$) down to the stratopause and below during polar winter conditions. The idea is that since NO is copiously produced in the lower thermosphere by auroral and photo-electrons (Gerard and Barth, 1977; Barth et al., 1988; Siskind et al., 1989, 1990), if some fraction of this NO were to be transported to the middle atmosphere, it could play a role in the stratospheric ozone budget (Langematz and Tully, 2018). This would signify that energetic electron precipitation (EEP) in the upper atmosphere could affect middle atmospheric (and possibly lower atmospheric) (Seppala et al., 2009) chemistry and climate. That such upper-to-middle atmospheric coupling might occur was proposed many years ago by Solomon et al. (1982). They noted that the most likely place for this to occur would be in the winter polar regions where the mean circulation favors descent and also where photolysis would naturally be weak. Observational validation of this idea remained limited (although see Russell et al., 1984) until, starting in the mid-1990's, observations of NO (Siskind and Russell, 1996; Siskind et al., 2000), $NO_2$ (Randall et al., 1998) and total odd nitrogen ($NO_y$) (Rinsland et al., 1999) provided evidence for the presence of upper atmospheric odd nitrogen at stratospheric altitudes. Since those early results, several other datasets, including the Michaelson Interferometer for Passive Atmospheric Sounding (MIPAS) (Funke et al., 2005; 2014), the Scanning Imaging Absorption Spectrometer for Atmospheric Cartography (SCIAMACHY) (Sinnhuber et al., 2016), the Atmospheric Chemistry Experient Fourier Transform Spectrometer (Bernath et al., 2005) (ACE-FTS, hereinafter just ACE) (Randall et al., 2007), the submillimeter radiometer (SMR) on the ODIN satellite (Perot et al., 2014), the Solar Occultation for Ice Experiment (SOFIE) on the Aeronomy of Ice in the Mesosphere (AIM) explorer (Bailey et al., 2014) and the Ozone Monitoring Instrument (OMI) on the NASA AURA satellite (Gordon et al., 2020) have contributed observations of this mechanism of thermosphere/middle atmosphere coupling.

The transport of thermospheric NOx to the stratosphere is manifested quite differently in Northern and Southern Hemispheres. In the South, due to relatively quiescent middle atmosphere dynamics, it is a regular occurrence and upper stratospheric NOx is seen to vary in accordance with geomagnetic activity (Siskind et al., 2000; Randall et al., 2007; Funke et al., 2014a; Gordon et al., 2020). By contrast, in the more dynamically active NH, this long range vertical transport is much more episodic. Of particular interest, and the subject of this work, is the descent of upper mesospheric air that has been observed after major sudden stratospheric warmings associated with so-called elevated stratopause events. Elevated stratopause events are phenomena whereby the middle atmospheric temperature maximum, normally situated near 50 km, reforms at or above 80 km right after a sudden stratospheric warming (Manney et al., 2005; Siskind et al., 2007; Chandran et al.,2011,2013; McLandress et al., 2013; Limpasuvan et al., 2016). This jump in stratopause elevation is followed by an extended recovery phase characterized by a marked spinup of the polar night jet, and most importantly for the present study, the descent of air from the upper mesosphere to the 45-55 km region. In the past 20 years, elevated stratopause events associaed with enhanced NO descent have been identified and documented in 2004, 2006, 2009, 2012, 2013 and 2019 (see Perot and Orsolini, 2021 for a summary overview). For these cases, the recovery phase has seen the presence of descending tongues of either enhanced CO and/or NO, or depleted $H_2O$, dropping over 30 km (Randall et al., 2005a/, 2006, 2009; Natarajan et al., 2004; Hauchecorne et

al., 2007; Manney et al., 2005,2008, 2009a, 2009b; Siskind et al. 2007; Bailey et al., 2014; Perot et al., 2014; Paivarinta et al., 2016). Of course these events were not all equally intense or appropriately timed during the winter season to yield maximal NO descent (cf. discussion by Holt et al., 2013); for example some dry mesospheric air was seen to descend after a SSW in 2010 (Straub et al., 2012) but there was no observed descent of upper mesospheric NO (Perot et al., 2014). The 2013 event, which is the focus of this paper, was one of the strongest in terms of bringing down enhanced NO from the upper mesosphere to the upper stratosphere.

Ultimately, the challenge associated with these events is in accurately simulating this mechanism of long range vertical coupling by whole atmosphere models. This has proven difficult. Simulations of mesospheric and lower thermospheric (MLT) NOx descent using the Whole Atmosphere Community Climate Model (WACCM, or its extended version WACCMX) have greatly underestimated the amount of NOx. This underestimate has been reported both for the Southern Hemisphere (Pettit et al., 2019) and for the Northern Hemisphere (Randall et al., 2015; Orsolini et al., 2017). A similar deficit was also reported with the Hamburg model of Neutral and Ionized Atmosphere (HAMMONIA) (Meraner et al., 2016) when operated in a free-running mode (i.e. unconstrained by observations). A comprehensive overview of models of NO descent is given by Funke et al (2017). They concluded that "the magnitude of the simulated NO tongue is generally underestimated by these models". This underestimate has raised questions about whether the deficiency lies in the neglect of photochemical sources from, for example, medium energy electrons or from an incomplete specification of mesospheric dynamics. In our previous work (Siskind et al., 2015), we supported the latter hypothesis, at least in years with low geomagnetic activity. Specifically, we showed that nudging WACCM with a meteorological analysis that extended up to the upper mesosphere and which presumably provided more realistic dynamical fields, yielded dramatic improvements in the representation of NO descent from the MLT to the lower mesosphere (Siskind et al., 2015). Pedatella et al. (2018) reached a similar conclusion, although their combined WACCMX/DART system still underestimated the descent.

Here, we follow up on those works to consider in more detail the ability of such a nudged model to capture this descent accurately. Our main focus is on how to characterize and quantify the delivery of MLT NO to the stratopause region, what we term the net deposition. As part of our analysis, we consider the two and three dimensional manifestation of the descent; most previous modeling studies have been limited to coarser averages over longitude and latitude (although see Salmi et al., 2011). In addition, our simulations, combined with other recent work in this area, have implications for the role of in-situ ionization occurring below 80-85 km during the late winter of 2013. Section 2 below discusses the modeling approach, Section 3 shows daily averaged results for specific latitudes, Section 4 presents results as a function of both latitude and longitude, Section 5 attempts to quantify the deposition of MLT NO in the upper stratosphere both as observed and as simulated by WACCMX and Section 6 discusses the implication of these results and provides conclusions.

## 2 Model Calculations

### 2.1 WACCMX

In this paper, we use the Whole Atmosphere Community Climate Model, extended version (WACCMX; Liu et al., 2010). The domain of this model extends from the ground to about 500 km. It is divided into 108 vertical levels such that the vertical resolution is variable around a quarter of the pressure scale height in the mesosphere and lower thermosphere; for stability and efficiency reasons the resolution asymptotes to half the local scale height in the thermosphere. WACCMX can be configured to use atmospheric specifications to constrain its meteorology (winds and temperature) from the ground to any altitude; this model configuration is referred to as Specified Dynamics (SD); see Sassi et al., (2013) for some details of the initial implementation. Note, the NO descent study of Siskind et al. (2015), to which we referred above, used WACCM, not WACCMX. More recently, McDonald et al. (2018) have documented significant improvements to the representation of tidal amplitudes when 3-hourly meteorological fields from NAVGEM-HA (see next section) are supplied to WACCMX. These are used to nudge the WACCMX winds and temperatures on a time scale of 1 hour. This short time scale ensures that not just the slowly varying fields are tied to the driving meteorology but also the shorter variability (like tides) is representative of the meteorology. It also ensures that the dynamical features in the large scale flow, such as, for example, the overall vortex structure, are essentially identical in both WACCMX and NAVGEM-HA. Sassi et al. (2018), using atmospheric specifications from NAVGEM-HA up to 90 km altitude in SD-WACCMX, showed a significant influence on the wave driving of the thermospheric circulation compared with using meteorological input only up to lower altitudes (e.g. 0-50 km). The simulation used in this study is exactly the same "hybma" (hybrid data assimilation with middle atmospheric observations) model run described in Sassi et al., (2021) that uses the NAVGEM-HA output. We note that unlike Hendrickx et al. (2018) and Orsolini et al., (2017) who studied NO transport into the upper mesosphere, our simulations use the standard eddy diffusion with Prandtl (Pr) number = 4. As these authors show, using Pr=2 (lower Pr increases the tracer diffusion) would lead to more rapid transport into the upper mesosphere. Specifically, Hendrickx et al (2018) showed that over a 2 week period, lower Pr causes NO to descend about 5 km lower in altitude. However, as we will discuss below, our upper mesospheric transport is already fairly rapid and decreasing Pr would likely worsen our agreement with SOFIE.

As in the previous studies referred to above, the nudging is applied to the WACCMX winds and temperatures, but not the trace constituents. As we will show, this means that simply because the large scale dynamical fields are identical in the model and the analysis, it does not guarantee that the tracer fields in WACCMX will respond identically to that given by NAVGEM. The model was started on Dec 1 from stabilized initial conditions generated by a previous run. The tracers are thus initialized at that time. Nitric oxide in the model is generated using the same auroral electron pattern used by Pedatella et al. (2018), namely, a fixed characteristic energy of 2 keV.

### 2.2 NAVGEM-HA

To constrain WACCMX dynamical fields, we use a meteorological analysis of winds, temperatures and constituents from a high altitude version of the Navy Global Envirnomental Model (NAVGEM-HA). NAVGEM-HA is the middle atmospheric

extension of the Navy's operational weather forecast system (Hogan et al., 2014). Details of the high altitude extension are provided by Eckermann et al., (2018); McCormack et al (2017) and Hoppel et al., (2013). Briefly, to supplement the operational troposheric and stratospheric observations used in the operational forecast system, middle atmosphere conditions in NAVGEM-HA are constrained by the additional assimilation of three satellite datasets. These include (1) Microwave Limb Sounder (MLS) temperature, ozone and water vapor (Schwartz et al., 2008) (2) temperature profiles from the Sounding of the Atmosphere using Broadband Emission Radiometry (SABER) (Remsberg et al., 2008; Rezac et al., 2015) and (3) microwave radiances from the upper atmosphere sounding channels of the Special Sensor Microwave Imager/Sounder (SSMIS) on the Defense Meteeteorological Satellite Program (DMSP) platforms (Swadley et al., 2008). Synoptic analyses of horizontal winds and temperatures are produced at a 6 hourly cadence and are used to initialize short-term forecasts, ultimately producing a 3-hourly analysis/forecast product each day up to $\approx$ 100 km. Validation of NAVGEM-HA wind fields against independent ground based wind measurements have been provided by McCormack et al.,(2017), Eckermann et al. (2018) and Jones et al. (2020). Also the tides derived from these winds have been compared with independent satellite (Dhadly et al., 2018) and ground based data (Stober et al., 2020). Finally, Eckermann et al. (2018) successfully compared NAVGEM-HA temperatures against AIM SOFIE data. However, there has not yet been published comparison of NAVGEM-HA tracer (i.e. $H_2O$) analyses with independent observations; this will be provided in Section 3 below.

## 3    Model Results

Our baseline comparisons are shown in Figures 1-4, which present model results that can be compared with previously published observations and simulations for the 2013 event. Figure 1 is a comparison of SOFIE NO and $H_2O$ data with calculated WACCMX results for the first 3 months of 2013. A similar figure of SOFIE data was first published by Bailey et al (2014, see their Figure 1), although here, in order to facilitate subsequent model comparisons, we show the SOFIE data on a pressure grid. Figure 1 shows zonal and daily averages of the SOFIE observations and of the corresponding model output at the local times and locations of the SOFIE occultations (i.e. local sunset, spacecraft sunrise). As we shall see, there is considerable longitudinal variability in the data. This means that a zonal average NO value equal to, for example, 100-200 ppbv actually reflects longitude sectors where much greater values are observed combined with many longitudes where the data were too low to allow for a retrieval. When the data are too low to allow for a retrieval, the mixing ratios are set to $10^{-5}$ ppbv. The top of the left hand column shows the progression of the occultation latitudes and the model is sampled at the same latitudes.

As seen in Figure 1, both observations and model have a tongue of enhanced NO descending downwards during this period. Looking at the detailed time evolution in all four panels, we see that initially the tracer fields are transported upwards in the week immediately after the warming (onset is Day 5 as per Orsolini et al., 2017). This uplift is seen in both the $H_2O$ and NO (although perhaps less so in the SOFIE $H_2O$ for reasons that are not clear; certainly the reduced sensitivity of SOFIE to such low mixing ratios above 80 km must be considered) and is consistent with similar initial uplifts simulated by Limpasuvan et al. (2016) and Orsolini et al. (2017). Figure 2 shows this more clearly by presenting the time evolution of the daily averaged altitude of the enhanced NO tongue in both WACCMX and SOFIE as defined by the pressure level of the 50 ppbv value. The

figure shows that in WACCMX, before the warming, 50 ppbv of NO are present as low as 0.04 hPa (about 70 km) and, in the week after the onset are brought up to 0.004 hPa (about 84 km for this date). Then, after about 5-10 days from the SSW, a strong persistent descent begins such that tracer values at 0.004 hPa are brought down to about 0.2 hPa (specifically, 0.18 hPa on our grid) by early February (Day 36). SOFIE shows the same behavior although the initial uplift appears somewhat more muted. The 50 ppbv marker in SOFIE starts at 0.01 hPa, rises to 0.004 hPa and then descends to 0.2 hPa by Day 45.

Figures 1 and 2 illustrate several points that are relevant for the rest of the paper. First, at least when viewed in the zonal mean, there is no evidence in either SOFIE or WACCMX for significant NO contributions to the middle and lower mesosphere from altitudes above 0.004 hPa, and even that pressure just represents air that was temporarily lifted upwards. This is generally consistent with Randall et al. (2001) who concluded that the immediate source region of the enhanced NO seen in the lower mesosphere and stratosphere was from the upper mesosphere and not the thermosphere. We have furthermore confirmed that this zonal mean perspective is representative by looking at individual longitudes in both SOFIE and WACCMX output. Thus in the SOFIE data, peak values of 1000 ppbv or greater are found at 0.004 hPa for specific longitudes in early January, are briefly uplifted to 0.002 hPa before descending, but never descend below 0.02 hPa. Likewise in WACCMX, the tongue of descending NO seen in Figure 1 is essentially peeled of from the bottom side of the NO layer. As a result of this analysis and in the absence of any contribution of NO from above 0.004 hPa, in subsequent discussion, we refer to this as upper mesospheric NO, not MLT NO.

This is important because while there are clearly differences between SOFIE and WACCMX between 0.004 and 0.001 hPa during this period (see Hendrickx et al., (2018) for more detailed comparisons of WACCM and SOFIE above 90 km), it is our assessment that these differences are not relevant to the present study. Second, the generally continuous variation in SOFIE shows no evidence for any production by in-situ ionization from energetic electrons as was observed recently by Duderstadt et al. (2021). Duderstadt et al. suggested that perhaps SOFIE sampled too far poleward to observe effects from radiation belt electrons. In any event, this speaks to a long standing question about the inability of WACCMX (or just WACCM) to deliver sufficient enhanced NO to the middle atmosphere (cf. Pedatella et al., 2018; Randall et al., 2015); our results suggest that for this event, transport from the 80-85 km region, not in-situ production at lower altitudes, led to the observed NO enhancements. Finally, it is clear that although NO in WACCMX reaches the mid-mesosphere about a week earlier than is observed, the descent in WACCMX appears to stall at that point. Thus the WACCMX 50 ppbv isoline reaches 0.18 hPa in early February and then only descends to 0.32 hPa over the course of the next month. By contrast, the observed NO keeps descending continuously and reaches about 0.75 hPa. This raises the question as to whether WACCMX can actually deliver upper mesospheric NO to the upper stratosphere where it might ultimately be important for ozone chemistry. As we will discuss in subsequent sections of this paper, to address this question the model-measurement comparison needs to be extended to two and three dimensions.

The differences in descent rate are also seen by tracking the evolution of the vertical profiles of $H_2O$ from SOFIE, WACCMX and also the NAVGEM-HA analysis, all sampled at the SOFIE occultation latitudes. This is seen in Figure 3, which presents zonal average $H_2O$ profiles for 15 day intervals from Jan 30 to March 15 (note: $H_2O$ fields prior to January 30th were not saved). The good agreement between NAVGEM-HA $H_2O$ and the independent SOFIE measurements serves as validation of the NAVGEM-HA assimilation of MLS data. Note WACCMX is only nudged to NAVGEM-HA dynamical fields, not tracer

fields and as noted above, WACCMX was initialized separately from NAVGEM-HA and SOFIE. Thus the mesospheric $H_2O$ in WACCMX is consistently higher than SOFIE and NAVGEM-HA. Nonetheless, we see important similarities in the shape of the profiles. On February 15, the biteout in the WACCMX profile and the local minima in the NAVGEM-HA and SOFIE profiles mark the location of the descending mesospheric dry layer at 0.1 hPa seen in Figure 1. This feature is seen descending in altitude on March 1 and March 15; however, it does not descend equally in WACCMX as compared to the analysis or the data. Thus in NAVGEM-HA, on March 1, the biteout appears at around 0.3-0.4 hPa, while the same feature in WACCMX remains at lower pressure, closer to 0.2 hPa. By March 15, the low $H_2O$ features in the 3 curves have all descended about a factor of 2 in pressure (to 0.6-0.7 in SOFIE/NAVGEM and to 0.4 hPa in WACCM) but because NAVGEM-HA and SOFIE were lower in altitude on March 1, they remain lower in altitude on March 15. Thus of interest is what happened between February 15 and March 15 that caused WACCMX to differ from the analysis and SOFIE. Another feature of interest is the increase in $H_2O$ above the biteout. As we will show in the next section, this is likely due to meridional transport.

We conclude this section with our other baseline comparison shown in Figure 4. This shows high latitude averaged mixing ratio profiles of NO for two specific dates, 19 Feb and 5 March, corresponding to the dates given in Figure 3 of Orsolini et al., 2017 (note, they referenced their profiles by the onset of the SSW on January 5 as Day 0; those day numbers are shown in the Figure). They compared WACCM simulations nudged by MERRA with ODIN/SMR data and showed that their calculated NO descent underpredicted the observations by over an order of magnitude. They show the SMR data as clearly presenting a tongue of enhanced NO descending throughout the period, but their model shows no evidence for such an enhancement. While we do not attempt to make a detailed comparison with the SMR data, we do present a few SMR data points taken from their figure to highlight the point made above, namely, that the WACCMX NO tongue appears to be displaced slightly higher in altitude than what is observed by SMR. Thus for Feb 19th, ODIN shows the peak at 0.2 hPa; the WACCMX peak is at 0.1 hPa. Likewise on March 5th, ODIN has the peak at 0.4 hPa; the WACCMX peak is at 0.2 hPa. This appears consistent with the stalling in the WACCMX descent shown in the comparison with SOFIE discussed above.

## 4  Two and 3D patterns of polar tracer descent

As discussed above, to answer the question as to whether WACCMX is delivering NO to the upper stratosphere, we need to look at the details of the transport in two and three dimensions. Figure 5 shows a series of pressure-latitude contour plots of daily and zonally averaged $H_2O$ for WACCMX (top row, panels a-d) and NAVGEM-HA (bottom row, panels e-h) at two week intervals from February 1 to March 15. In each panel, two reference lines are indicated. First the vertical dashed lines reflect the latitude of the SOFIE occultation measurement as previously indicated in Figures 1 and 3. Second, a reference horizontal line at approximately 0.32 hPa is highlighted; this pressure will be examined more closely in figures below.

It is immediately clear that in the WACCMX results for February 1 and 15, the strongest descent occurs not at the pole but near 70°N, i.e. where a noticeable dip in the contour lines is seen. By March 1st, the sub polar descent in WACCMX has faded and there is a suggestion that the overall descent has noticeably weakened, i.e. the very lowest values of $H_2O$ have receded upwards somewhat (look at the pressure level associated with the 5.0 ppmv contour). This weakening of the descent may be

at least partially consistent with the apparent stalling of the descent we discussed in the previous section. A second reason for this apparent stalling of descent in WACCMX is the sampling. With a postive gradient of model $H_2O$ going from sub-polar to polar latitudes, as the SOFIE occultation latitudes move to higher latitudes, it will tend to sample model regions with somewhat higher values of $H_2O$. Thus we see that on February 15, the SOFIE sampling has moved poleward of the minimum value of

5 the $H_2O$ in the lower mesosphere. This will be also be seen clearly below when we show polar plots.

  In the bottom row of Figure 5 are the corresonding results from NAVGEM-HA. We see both some interesting similarities and differences. What is similar is that NAVGEM-HA does show evidence for off-polar descent on February 1st, and particularly on February 15th. This can be seen in the slope of the isolines crossing the 0.32 hPa fiducial line whereby the contour lines show a poleward and upwards tilt from 65 to 85°N and thus the minimum value on February 15 is, like WACCMX, near 70N.

By March 1, however, all semblance of off-polar descent in the NAVGEM-HA field is gone and strong pole-centered descent is now quite clear. As we saw in the profile comparison with SOFIE, the tongue of low $H_2O$ (defined qualitatively by the 4-5 ppmv contour) penetrates down to the 0.4-0.5 hPa level, whereas it remains at 0.2-0.3 hPa in WACCMX. Finally, one area of similarity between NAVGEM-HA and WACCMX is at higher altitudes, between 0.01 and 0.1 hPa, where both the analysis and the model show evidence for wetter air moving poleward. Certainly, the absolute values are different but, as noted above,

this may reflect the initialization of WACCMX in December from climatology. The poleward motion evidenced by both the model and the analysis suggests some sort of overturning above the descending air and this is clearly seen in the NAVGEM-HA analysis by March 15. This is consistent with the SOFIE data shown in Figure 1 which shows higher values of $H_2O$ above the tongues of descending air.

  The existence and consequence of off-polar descent, most notably in WACCM but also to some degree in NAVGEM-HA as

well, can be dramatically visualized in the polar plots presented in Figure 6. These are surfaces of $H_2O$ mixing ratio, for the first 3 dates shown in Figure 5, at the 0.32 hPa level that was shown as a reference line in Figure 5. Also shown as thick black rings, are the SOFIE occultation latitudes for these dates. The off-polar descent is manifested as a ring of circumpolar dry air, with a local mixing ratio maximum centered at the pole. This is present in WACCMX on February 1st and in both WACCMX and NAVGEM-HA on February 15, but dissolves in NAVGEM-HA by March 01. The effect of the poleward progression of

the SOFIE sampling in WACCMX can been seen by the fact that the smaller black ring in the WACCMX February 15 panel completely misses the dry annulus of $H_2O$ even though the actual value of $H_2O$ in this annulus is lower on February 15 than on February 1st. Therefore we see that descent has occurred in WACCMX, but the SOFIE sampling completely misses it and continues to sample only air with mixing ratios $> 5$ ppmv (the yellow colors). This demonstrates the need for care in sampling general circulation models to correspond to occultation data at single latitudes.

By March 1st, the WACCMX and NAVGEM-HA fields are very different. Thus the annulus in WACCMX is still present, but displaced such that the local maximum is over Northern Greenland. By contrast, these longitudes are precisely where the deep minimum in NAVGEM-HA $H_2O$ is seen. The relative phasing of WACCMX and NAVGEM-HA $H_2O$ and temperature can be most clearly seen in Figure 7 which presents line plots of the longitudinal variability of these WACCMX and NAVGEM-HA fields, and additionally, shows the SOFIE data for comparison. On February 15, in general, all three temperature products agree;

there is a small wave with a maximum near 80-100°E and a minimum near 220-240°E. The $H_2O$ fields show all little variation

for this date (the absolute abundance in WACCMX is higher, consistent with what was discussed in regards to Figure 4). By March 1, the situation is different. The temperature variation is greater for all three products and there are some interesting differences in the water vapor variation. The WACCMX field shows a general anticorrelation between the longitudinal variation in $H_2O$ and temperature while NAVGEM-HA and SOFIE show a positive correlation. These differences are, at first glance,

confusing. In general we'd expect the downward advection of dry mesospheric air to be associated with the descending warm stratopause, i.e. an anticorrelation, consistent with observations (Manney et al., 2008). WACCMX conforms to this expectation but NAVGEM-HA and SOFIE do not and are almost 180 degrees out of phase- they show the lowest $H_2O$ corresponding to the coldest temperatures. Here, having the SOFIE data as an independent validation is useful for what otherwise would be a puzzling difference between WACCMX and NAVGEM-HA.

We suggest that the differences between WACCMX and NAVGEM-HA are likely due to differences in the relative roles of vertical and horizontal advection. We can gain some insights into this by looking at the Transformed Eulerian Mean (TEM) circulations inferred from the WACCMX and NAVGEM-HA wind fields. This can be seen in Figure 8 which compares the monthly averaged (for February) TEM horizontal and vertical winds calculated from WACCMX and NAVGEM-HA for the mesosphere plotted poleward of 30°N. Following Siskind et al. (2010), our approach is to directly evaluate the expression

(equation 3.5.1a from Andrews et al. (1987))

$$\overline{v^*} = v - \rho_o^{-1}(\rho_o \overline{v'\theta'}/\overline{\theta}_z)_z \tag{1}$$

and then solve for $\overline{w^*}$ from the continuity equation (equation 3.5.2c of Andrews et al., 1987).

Between 0.1 and 0.01 hPa, both WACCMX and NAVGEM-HA are in general similar. They both show broad poleward and downward flow. This poleward flow, which is in general well known from previous work (e.g. Smith et al., 2011) explains the

20 increase in $H_2O$ at these altitudes noted above as being due to the horizontal advection of wetter air. Between 1.0 and 0.1 hPa, the flow weakens in both WACCMX and NAVGEM-HA and here some important differences emerge. Thus in WACCMX, the descent becomes very weak poleward of 70N and indeed at around 0.2-0.3 hPa, a zero isoline is seen, below which w* is positive, implying ascent. In WACCMX, the only place where descent continues unabated to 1.0 hPa is around 70°N, which closely coincides with the dry annulus seen in the calculated $H_2O$ shown in Figures 5 and 6. Note, when we compare with

25 SOFIE, we must compare at the latitudes of the SOFIE occulation. By late February, as seen in Figure 1, this is poleward of 75N. Thus the stalling out of the simulated descent at SOFIE latitudes is consistent with this zero line in the descent. By contrast, in NAVGEM-HA, w* is negative at all altitudes for mid-to-high latitudes and in the lower mesosphere.

Regarding the meridional flow, between 0.1 and 0.2 hPa poleward of 70N, there is a layer of equatorward flow in both WACCMX and NAVGEM-HA. Thus in WACCMX air initially centered over the pole would descend to 0.2 hPa, move equa-

30 torward and then continue descending at 70N. This would produce the annular patterns. But below 0.2 hPa, WACCMX and NAVGEM-HA differ. Thus while at 0.1-0.2 hPa, equatorward flow is seen in NAVGEM-HA, at immediately higher pressures, it then reverses such that below 0.3 hPa the flow is poleward again. Thus the 0.3 hPa level, at least in February, appears to be a transition level between off polar descent reversing to polar-centered descent. As a result, we see the analyzed $H_2O$ go from a

pattern where the minimum values appear as an annulus on Feb 15th to one where the minimum values are more pole-centered (although offset somewhat towards Greenland)

In general, the residual mean circulation is driven by wave forcing, either from large scale planetary waves or small scale gravity waves (Andrews et al., 1987; Smith, 2012). Since the larger scale forcing is presumably constrained by NAVGEM-HA, any differences in v* and w* between WACCMX and NAVGEM-HA are most likely due to unresolved small scale gravity waves that are not exactly captured either by the nudging procedure or by the WACCMX gravity wave parameterization. A truly comprehensive examination of the causes of these differences is beyond the scope of the present study. The differences in the distribution of $H_2O$ do suggest that nudging to realistic meteorology, while a necessary step, may in and of itself be insufficient to capture the details of mesospheric descent in the polar vortex. Further, as we will discuss in the next section, the persistence of an annular descent pattern in WACCMX complicates the comparison of WACCMX NO and occultation data such as SOFIE. The differences between WACCMX and SOFIE in the $H_2O$ variation will have analogs in our model-data NO comparison. However, despite these differences, we will show that the overall net transport of NO in WACCMX down to the stratopause is probably not far off from that suggested by the observations.

## 5    Delivery of upper mesospheric NO into the stratosphere

Here we attempt to evaluate the net deposition of upper mesospheric WACCMX NOx into the stratosphere and compare the model with satellite observations. In doing this comparison we must take into account the complex WACCMX tracer descent pattern outlined above, whereby vortex-centered descent appears to encounter a level where w* goes to zero and the descent thus diverges into an annulus. This pattern of descent in WACCMX means that a straightforward comparison of WACCMX with satellite data can be misleading. The satellite data that we use to illustrate this are from SOFIE and ACE. ACE is similar to SOFIE in that it is an occultation experiment; however, its orbit differs from AIM so the latitudinal sampling is different. Bailey et al. [2014, figure 1] show the latitudes sampled from both of the experiments. They show that near the beginning of March, they are both sampling near $81^{\circ}$N and diverge as the month progresses. We will perform our comparisons for March 1-3, when ACE and SOFIE were roughly coincident and also for the period March 23-25 when ACE was sampling around 68-72$^{\circ}$N and SOFIE was near the pole (87-88$^{\circ}$N). After this time ACE moved too far south to observe enhanced polar NO.

Figure 9 shows an overall comparison of NO altitude profiles from SOFIE and ACE for the two periods we consider here. Since both instruments are observing the same latitude for March 1-3, the left panel can be considered the more valid intercomparison. Both data products are provided on a very fine altitude grid (1 km for ACE, 200 meters for SOFIE) and the plot reflects that; however, their native vertical resolutions are coarser and also different for each experiment (3.5 km for ACE and 2.5 km for SOFIE (Hervig et al., 2019). This appears to show up in the figure where it seems that the NO layer recorded by SOFIE is narrower and perhaps peaked slightly more sharply than in ACE. With this consideration in mind, the intercomparison for March 1-3 shows good qualitative agreement. For example, both instruments show descent from peaks centered near 0.32 hPa in early March descending to about 1 hPa for March 23-25. We will present an additional comparison below using vortex

centered coordinates (equivalent latitudes) which will provide useful additional context for the March 23-25 comparison when the two instruments were not sampling the same latitude.

Figure 10 presents the longitudinal variation of the observations compared with WACCMX, sampled similarly, for two pressure levels, for the beginning of March when ACE and AIM are sampling the same latitude. The left panel, 0.24 hPa, corresponds to the approximate peak of the WACCMX NO tongue as seen in Figures 1 and 2. The right panel, 0.42 hPa, corresponds to the WACCMX level closest to the approximate peak of the SOFIE and ACE data shown in Figure 9. It is important to note that both ACE and SOFIE sampled all longitudes; however, at those longitudes where the signal is too low to allow a meaningful retrieval, values of either $10^{-14}$ or negative values are seen in the databases. These appear in the figure as the symbols near zero. Below the model data comparison, we show a dynamical indicator, the equivalent latitude. This is calculated using 3-day average potential vorticity fields from WACCMX. The averaging in time acted to smooth some of the complex PV spatial structure that occurs in the mesosphere (e.g. Harvey et al., 2009).

The use of equivalent latitude is important because it places the data and the model in a vortex centered framework. It shows that the equivalent latitude can vary by a significant amount as a function of longitude around a single latitude circle (e.g. see Randall et al., 2002; Randall et al., 2005a). This means that even though ACE and SOFIE are only sampling a single latitude on a given day, they actually are sampling different regions of the vortex. This will be helpful in comparing the satellite data to the model. Both ACE and AIM tend to maximize at equivalent latitudes greater than 75-80 (i.e. at longitudes less than 30 or greater than 200-220). Note that these are the same longitudes that NAVGEM-HA and SOFIE showed the lowest $H_2O$ in Figure 7- i.e. descent is maximum here. At 0.24 hPa, the longitudinal variation in WACCMX NO tends to mirror the observations; however, at 0.42 hPa, WACCMX is quite different- showing minimum NO at precisely the same longitudes where the observations show maxima. Conversely, it shows elevated NO values at the lower equivalent latitudes where the observations fail to record any data (i.e. longitudes from 80-200). This near out-of-phase behavior of the WACCMX NO is analogous to what was seen in the WACCMX water vapor in Figure 7. Thus the data show maximal descent with concomitant NO enhancement inside the vortex near 270-330E, but the model shows descent away from the vortex core. It is interesting to note that the location of the vortex as shown in Figure 6 and NO descent maximum as shown here, both near Greenland, is very similar as observed for other extended SSW events such as in 2004 (Winick et al., 2009) and 2009 (Harvey et al., 2021). We next quantify the overall global, total amount of upper mesospheric NO that has descended to the upper stratosphere/lower mesosphere (net deposition). To compensate for the detailed differences in WACCMX and the observations, we will use three different approaches. None of these are foolproof; however, taken in aggregate, they show that we can place a general bound on the net deposition upper mesospheric NO into the lower mesosphere/upper stratosphere region, despite the fact that the NO enhancement is not necessarily manifest in the same geographic locations in WACCMX as it is in the observations. Figure 11 presents a global polar cap overview of the relevant quantities we used in our analysis. The figure shows polar stereographic views of equivalent latitude, geopotential height, $CH_4$ mixing ratio (ppmv), and $NO_x$ mixing ratio ($\log_{10}$(ppbv)). Note, Figure 11 shows $NO_x$ (=NO + $NO_2$), not simply NO as in Figures 9 and 10. This is because NO has a pronounced diurnal variation whereby it forms $NO_2$ at night and is reformed rapidly each day due to $NO_2$ photolysis at visible wavelengths. On the other hand, $NO_x$ is much longer lived (the discussion regarding Figure 15 below will quantify this in more detail) and is more useful

as a dynamical tracer, as it is presented in Figure 11. During daylight hours, for the pressures shown here, the difference between NO and $NO_x$ is small ($< 5\%$) and we have confirmed this by looking both at the ACE $NO_2$ data (SOFIE does not measure $NO_2$ so cannot be used for this specific assessment) as well as the WACCM NO and $NO_2$ fields. Thus when we evaluate the total WACCMX budget of $NO_x$, this will differ little from the total NO budget. Finally, on all panels, the thick black line

indicates the vortex edge as defined by the method described by Harvey et al (2002) which identifies the streamfunction contour coincident with the polar night jet at each altitude. Note that the vortex edge is not precisely aligned with an equivalent latitude contour since streamfunction and potential vorticity contours (used for equivalent latitude) are not absolutely parallel.

   One of our approaches to quantifying net descent will be vortex centered, i.e. how much WACCMX $NO_x$ is contained inside the polar vortex defined by the black line. A second approach will account for the fact that, as seen in Figure 9, and evident here,

the variation of calculated $NO_x$ doesn't always appear well correlated with the location of the vortex. Thus at 0.24 hPa, while the enhanced NOx is generally entrained inside the vortex, the crescent shaped distribution of the values greater than 100 ppbv (2.0 on the plot) does not correspond to the vortex edge. More dramatically, at 0.42 hPa, the model pattern is annular, much as the $H_2O$ was seen to be in Figure 5. Further, over western North America at 0.42 hPa, where the $65°$ equivalent latitude contour is labeled, we see enhanced NOx (green values exceeding 1.4, i.e. mixing ratios greater than 25 ppbv), extending outside the

vortex, to and below $60°$ equivalent latitude. This is accompanied by lower values in the vortex core. To attempt to account for this varied dependence of $NO_x$ on equivalent latitude, we will evaluate the amount of WACCMX $NO_x$ that is bounded by different ranges of equivalent latitude. Finally, the third approach will be to use regions of low $CH_4$ as a flag for enhanced $NO_x$. This will also have complications as we discuss below. These three approaches to quantifying the descending NO in WACCMX will then be compared to a simple geometric model of the latitudinal variation of the NO based upon the SOFIE

and ACE data.

   A summary of these different results is presented in Table 1 where the total NOx is expressed in gigamoles (GM), for the two periods presented in Figure 9. As shown in Figure 9 (and also Figures 1 and 2), the observed NO has descended over the three week period considered; the pressure ranges for integrating the column NO in the model are approximately .18-.42 hPa (about 50-56 km) for March 1-3 and .42-1.15 hPa (about 43-50 km) for March 23-25. The result of the first approach,

i.e. to simply integrate over all the NOx in the polar vortex, is presented in column 1. This approach has the advantage of objectivity, the vortex edge is defined by the appropriate streamfunction gradient and we adopt that for the area to consider. The problem, as noted above, is that there is evidence, at least at some pressures, that enhanced NO has spilled out of the model vortex. At the same time, we may want to account for the possibility that, with the off-polar descent, enhanced NO may not be present in the vortex core. If we simply integrated all the NOx molecules over the entire polar vortex, under this

scenario, we might be including molecules that were not of upper mesospheric origin. Use of equivalent latitudes allows us to account for both possibilities. The problem with this approach is that it is hard to define a single objective criterion for which range of equivalent latitudes to use. Thus we select three bands to show the sensitivity of the answer to the different ranges-these are given in columns 2-4 of the Table. In general, the values obtained with this approach are roughly consistent with the vortex-only approach. The third and final approach to estimating the amount of upper mesospheric NO in WACCMX is to use

the anticorrelation between $CH_4$ and upper mesospheric NO. This has been long used in observational analyses, starting with

the Halogen Occultation Experiment (HALOE) data on the UARS satellite, (cf. Callis et al., 1996; Siskind and Russell, 1996; Siskind et al., 2000) and relies on the idea that in general, $CH_4$ and $NO_x$ should be positively correlated in the stratosphere since both are ultimately of tropospheric origin. However, for those situations where air from the upper mesosphere has descended to the stratopshere, $CH_4$ and $NO_x$ will be anticorrelated since upper mesospheric air is depleted in $CH_4$ but enriched in $NO_x$.

With HALOE data, there typically was a single value of $CH_4$ that could be used as a threshold to distinguish between the two air masses (cf Figure 4 of Siskind and Russell, 1996). The problem is that in WACCM, the $CH_4$-$NO_x$ relationship is not necessarily single valued and the correlation between $CH_4$ and $NO_x$ is often ambiguous.

The complexity of the WACCMX $CH_4$ and $NO_x$ relationship in WACCMX is shown in Figure 12 for two pressures that correspond to the location of the WACCMX NO layer in Figure 1 for March 1-3 and March 23-25. The horizontal dotted lines

in each panel are possible $CH_4$ thresholds below which the air might be assumed to be of upper mesospheric origin. The upper threshold includes more values of $CH_4$ to be considered as an indicator of upper mesospheric air and thus using this threshold yields larger values of upper mesospheric NOx. The lower threshold is more restrictive. But no single threshold works very well. For example, the 0.24 ppmv threshold at 0.42 hPa in the March 1-3 period includes a region near NOx values of 8-12 ppbv, where NOx and $CH_4$ appear to be both positively and negatively correlated. Likewise for the March 23-25th period,

there is a range of $CH_4$ values associated with NOx values of 10-12 ppbv with no clear slope. This has the effect of making it unclear whether NOx values in the range 10-12 ppbv are truly of upper mesospheric or of tropospheric origin. Thus using the upper threshold risks including lower atmospheric air in the estimate for upper mesospheric NOx and indeed our estimates with this threshold, shown in column 5 of Table 1, exceed the values from the vortex and equivalent latitude approaches. Funke et al. (2014b) discuss this problem in terms of different air parcels experiencing different photochemical histories, although the

examples they discuss have to do with seasonal effects that should be less important here. Regardless of the cause, to illustrate the sensitivity of our results, the 2nd more restrictive (i.e. lower) threshold for $CH_4$ yields an integrated upper mesospheric NO abundance that is lower and closer to the values obtained using equivalent latitude criteria. This is given in the sixth column of Table 1. Overall, our model estimates agree with the general range given by Funke et al. (2014a), especially if one recognizes that for several of the years in which they get larger values than we show in Table 1 (e.g. 2008) much of the transport occurs in

the earlier part of the winter that we do not evaluate here.

The various model-derived global NOx values in Table 1 can be compared with our geometric model estimates for the two periods that are, in turn, compared with ACE and SOFIE data. Our geometric approach to estimate the upper mesospheric NO abundance implied by the observations is different than with the model since we do not have complete global coverage. However, we can make some plausible estimates by looking at the variation of ACE and SOFIE data with equivalent latitude.

This is shown in Figure 13. The data in the left column can be compared to the WACCMX data seen in Figures 10 and 11. The data in the right column can be compared with Figure 14 which shows model results for March 23-25 for the three pressures which roughly bracket the enhanced NO in WACCMX seen in Figure 1. In general, for both dates, it appears that the measurements do not spread to such low equivalent latitudes as suggested by the model. Whereas Figure 11 indicated the presence of upper mesospheric NOx in WACCMX over the western United States at equivalent latitudes of 55-60, the

observations for March 1-3 do not record any NO at equivalent latitudes less than about 73° for this period (note, both AIM

and SOFIE were taking samples at these locations but if the signal is too low for a good retrieval, they will report near zero values in their databases). Likewise for the March 23-25 period, Figure 14 shows that the model NO often maximizes at equivalent latitudes between 65-70° while the data show clear fall offs for equivalent latitudes less than 70. Our specific approach to estimating how much NO might be consistent with the observations is to make the very simple assumption that it is constant as per the horizontal lines in the figure, over the indicated range of equivalent latitudes (i.e. down to 73N for March 1-3 and 70N for March 23-35), for a 5 km thick layer. The resulting estimates from this approach are given in the last column of Table 1. It is interesting that our estimates are of the same order as, albeit perhaps a factor of 1.5-2 lower than, our estimates from the model.

The advantage of this approach is that it is ultimately based upon simple geometry and potential uncertainties can be readily assessed. The two main uncertainties we address are the actual thickness of the layer and its latitudinal (specifically, equator-ward) extent. Regarding the thickness of the enhanced NO layer, the data suggest that the layer is narrow and near the limit of the vertical resolution of both the satellites and of WACCMX (recall WACCMX gridding corresponds to about 2 km resolution near the stratopause). With our geometric approach, we can easily scale our answers. Thus if the layer were actually 6 km thick, our estimates would be low by 20%; if the layer were 4 km thick, our estimates might be equivalently too high. Regarding the latitudinal extent, this is probably the larger uncertainty since the area of the globe increases as one moves equatorward. If we were to assume that 20 ppbv of NO extended to equivalent latitudes as low as 61N, this would double our estimated value for March 23-25 and would bring it more to the middle of the model estimates. Enhanced NO at such low equivalent latitudes does not appear to be present in the data and thus we give an approximate upper limit of a factor of 2 for the accuracy of our estimates.

Ultimately, all these comparisons and indirect approaches reinforce the same point, namely that upper mesospheric NO (or NOx) in WACCMX does not present itself at the same altitudes, latitudes or longitudes as indicated by the SOFIE and ACE data. However, the global totals are quite close. We conclude that WACCMX/NAVGEM-HA represents the NOx descent to the lower mesosphere/upper stratosphere reasonably well during this period, and no additional source of NOx is needed to reproduce the total amount of EPP-IE NOx during this time.

# 6   Discussion and Conclusions

We draw two conclusions from our study. The first deals with the 2 and 3D morphology of the mesospheric descent. It appears that subtle differences in the residual mean circulation, presumably due to equivalently subtle differences in gravity wave forcing, can have surprisingly large effects on tracer transport near the stratopause. The existence of off-polar descent was seen many years ago in analyses of the Antarctic polar vortex (Manney et al., 1994); our results here, reflecting the MLS observations assimilated into NAVGEM-HA, provide the first observational support for this morphology at lower mesospheric altitudes. At the same time, it is clear that this effect is overestimated in WACCMX. Another consequence of the differences in tracer transport between WACCMX and observations as discussed in the context of Figures 8, 13, and 14 is that upper mesospheric NOx in WACCMX is advected equatorward more than what ACE and SOFIE suggest. This then could have some

consequences for the ability of WACCMX to incorporate upper mesospheric NOx into the stratospheric NOx layer; namely NO may photodissociate too quickly in the model. This is illustrated by Figure 15 which presents the diurnally averaged lifetime of NO against photodissociation using the parameterization of Minschwaner and Siskind, (1993) as a function of latitude for equinox conditions. It shows that for latitudes equatorward of 70°N, where much of the NO in WACCMX is found, the lifetime against dissociation is on the order of 2-4 weeks at 1 hPa. By contrast, for the regions where the NO in the observations is found, in the vortex core, poleward of 75°N, the lifetime is much longer- closer to 2-3 months. Nitric oxide here would see much less sunlight than in the model, and could continue descending lower into the stratosphere.

Our second main conclusion follows from our earlier study (Siskind et al., 2015) and is that we do not require any added in-situ chemical production from energetic electrons below 80-85 km. This conclusion echoes that reached by Shepherd et al., (2014) for the 2006 SSW event. To be fair, this period of time (late winter 2013) appears to not coincide with significant geomagnetic activity. Certainly, for cases with very large heliophysical forcing such as in Northern winter 2003-2004 (Randall et al., 2005b) or during solar proton events as in 2005 (Andersson et al., 2016), the situation is quite different. Our results also may not completely apply to the Southern Hemisphere where the dynamics are different and NO from EPP effects are routinely seen in the middle stratosphere. However, our results should provide a constraint for models that are used to conduct sensitivity studies of the effects of such ionization. Ultimately, our results continue to point to the need for accurate meteorological analyses which extend into the mesosphere, such as provided by NAVGEM-HA.

*Code and data availability.*  The NAVGEM-HA analysis data is accessible at https://map.nrl.navy.mil, cd to map/pub/nrl/jgrspace2020/lightspecies/navgem. The WACCMX model is open-source and available from https://www.acom.ucar.edu. SOFIE data is available from https://sofie.gats-inc.com. ACE data was obtained from https://databace.scisat.ca/level2/

*Author contributions.*  D. E. Siskind conceived the study, conducted most of the analysis and wrote most of the paper, V.L. Harvey performed the equivalent latitude analysis and edited the sections of the text which described it, F. Sassi provided the WACCMX simulations and edited the section which described WACCMX, J. P. McCormack provided the NAVGEM-HA analysis and edited the section which described it, C. E. Randall evaluated the present results in the context of past studies of medium energy electrons and edited relevant sections of the paper, M. E. Hervig is the PI of SOFIE and advised on interpretation of the SOFIE retrieval in the mesosphere, S. M. Bailey advised on the comparison of SOFIE to the model and helped edit the paper

*Competing interests.*  The authors declare that they have no competing interests

*Acknowledgements.* We acknowledge funding from the NASA Aeronomy of Ice in the Mesosphere explorer program. In addition, FS acknowledges funding from the Chief of Naval Research, JPM acknowledges funding from the NASA Living with a Star program under interagency agreement NNH13AV95I, and CER acknowledges funding from the NSF CEDAR program, grant ACS 1651428.

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

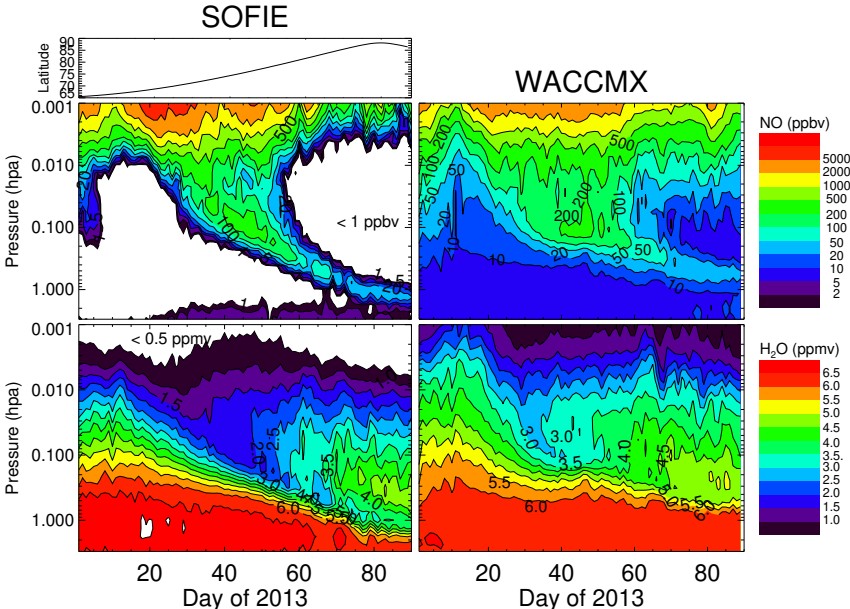

**Figure 1.** Comparison of the time evolution of daily zonal mean NO (top color panel, units of ppbv), and $H_2O$ (bottom, units of ppmv) from SOFIE (left column) and WACCMX (right column) for the first 88 days of 2013. The model is sampled according to the latitudes and longitudes tracked by the SOFIE occultation pattern. The latitudinal pattern is shown in the uppermost panel on the left column, going from about 66N on January 1 to about 88N at the March equinox.

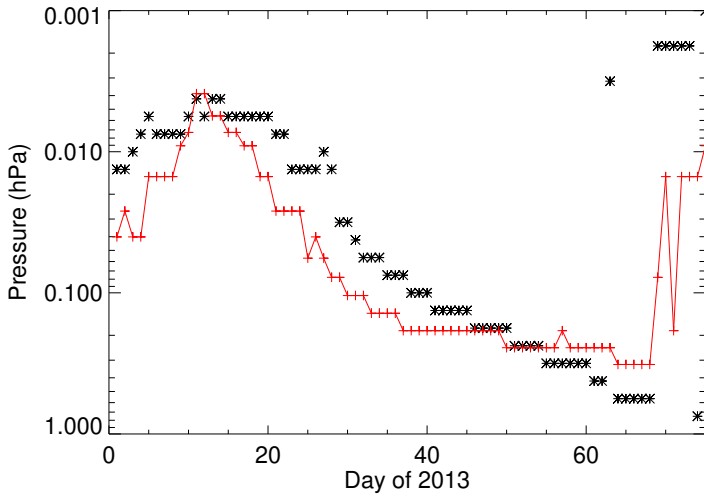

**Figure 2.** The pressure level of the nitric oxide 50 ppbv value (bottom side of the layer). The pluses connected by a red line are daily values of the WACCMX results shown in Figure 1. The black stars are the SOFIE data from Figure 1.

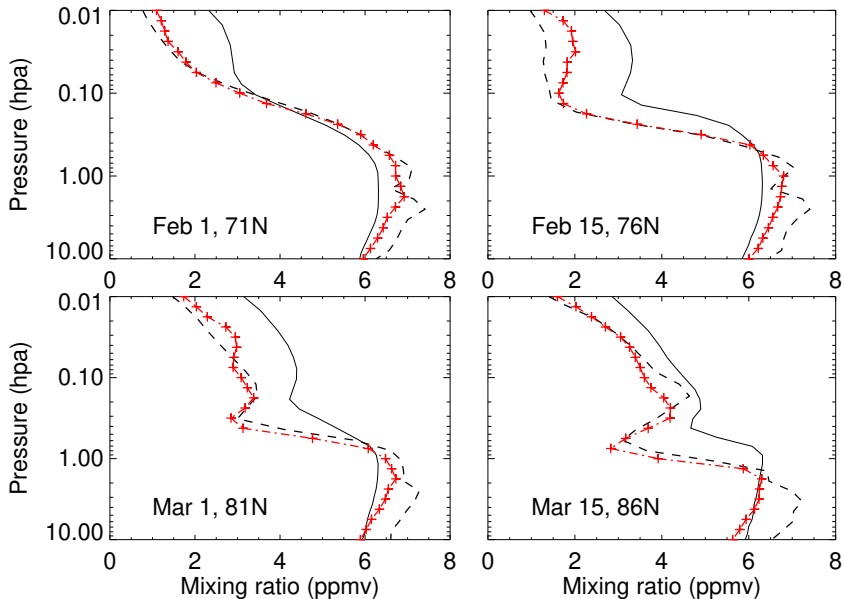

**Figure 3.** Zonally averaged H$_2$O profiles from (solid black) WACCMX calculations, (dashed black) NAVGEM-HA analysis and (red) SOFIE observations. Both WACCMX and NAVGEM-HA are sampled at the SOFIE occultation latitudes.

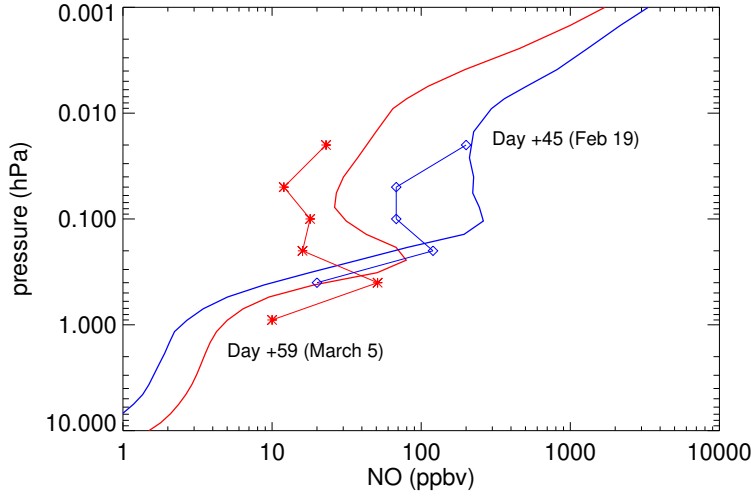

**Figure 4.** Calculated nitric oxide profiles, averaged from 70-90N, from WACCMX for the indicated dates to compare with Figure 3 of Orsolini et al., [2017]. The day numbers reflect the convention used by Orsolini et al. and are references to the SSW onset on January 5th (Day 0). The actual dates are shown for reference. The symbols are from the SMR data presented by Oroslini et al. (red stars for March 5; blue diamonds for Feb 19)

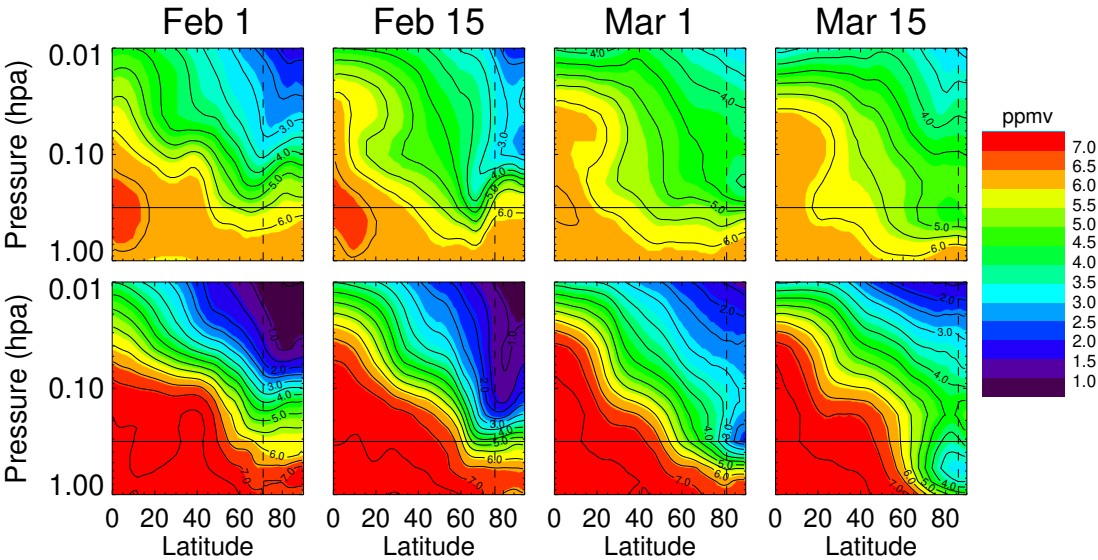

**Figure 5.** Daily and zonally averaged H$_2$O from WACCMX (top row) and NAVGEM-HA (bottom row) for the indicated dates. The horizontal line in each panel is a fiducial to mark the 0.32 hPa level. The vertical dashed lines mark the latitude of the SOFIE occultations for each date.

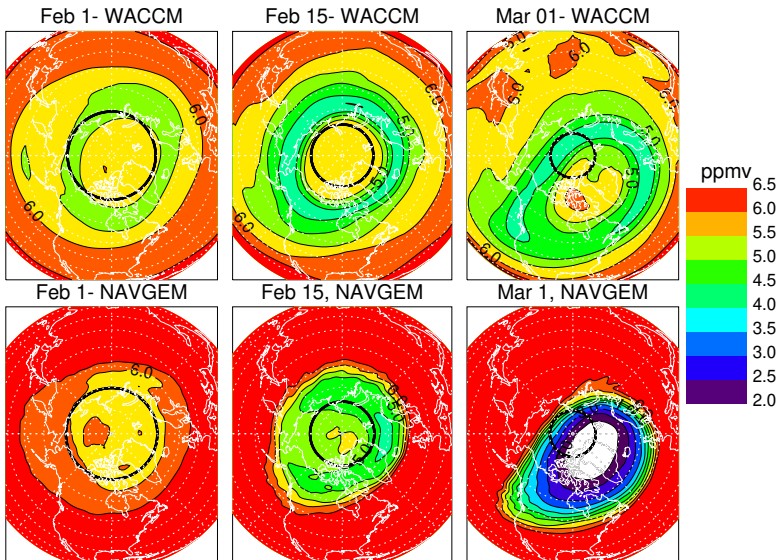

**Figure 6.** Polar projections at 0.32 hPa of $H_2O$ from WACCMX (top row) and NAVGEM-HA (bottom) for the indicated dates. The contour interval is 0.5 ppmv. The brightest red is greater than 6.5 ppmv and the white in the NAVGEM-HA field for March 1st represents values less than 2.0 ppmv. The black rings in each panel represent the occultation latitudes for SOFIE for each date. The polar plots are oriented such that a longitude of 90W is at the bottom.

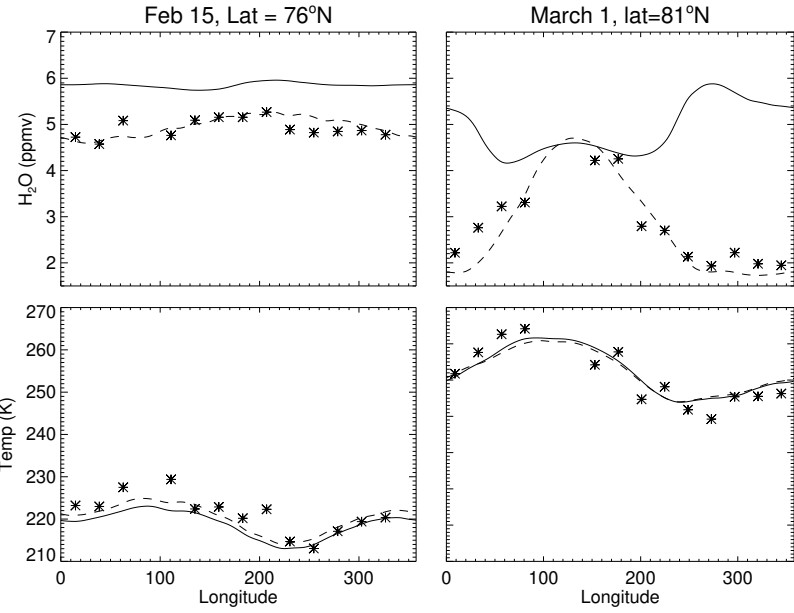

**Figure 7.** Calculated WACCMX (solid), analyzed NAVGEM-HA (dashed) and observed SOFIE (stars) water vapor (top row) and temperature (bottom row) for p = 0.32 hPa versus longitude for the indicated dates and latitudes.

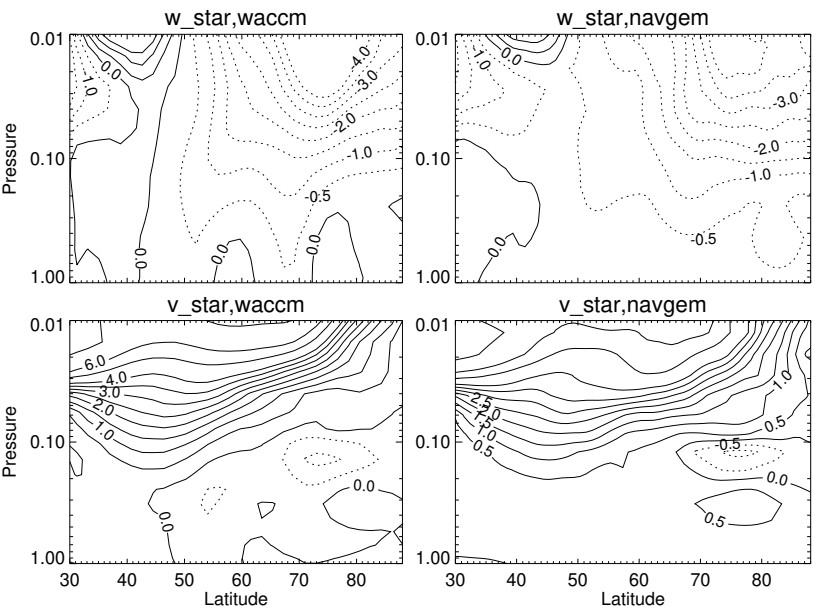

**Figure 8.** Comparison of monthly mean (February) w* (top row) and v* (bottom row) for WACCMX (left column) and NAVGEM-HA (right column). For w*, the units are cm/sec with a contour interval of 0.5 cm/sec and for v*, the units are m/sec and the contour interval is 0.5 m/sec. Negative values are dotted contours.

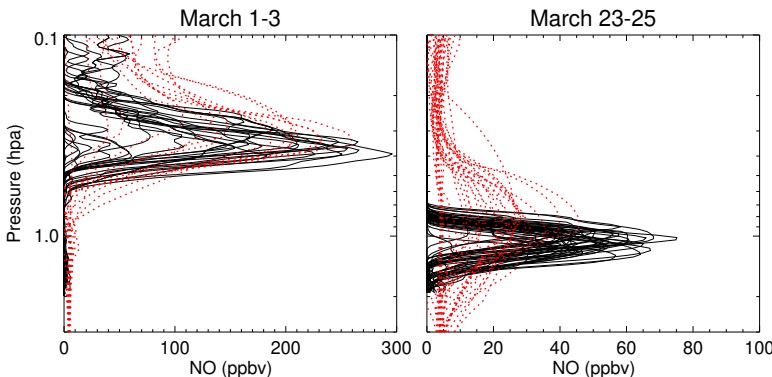

**Figure 9.** Comparison of nitric oxide data recorded by SOFIE (solid black curves) and ACE (dotted, red) for the two 3 day periods indicated. For March 1-3, both instruments are observing near 81N; for March 23-25, SOFIE is observing near 88N and ACE is observing in the range 68-72N. The consequences of SOFIE and ACE observing at these different latitudes for the 2nd date range are discussed in the text.

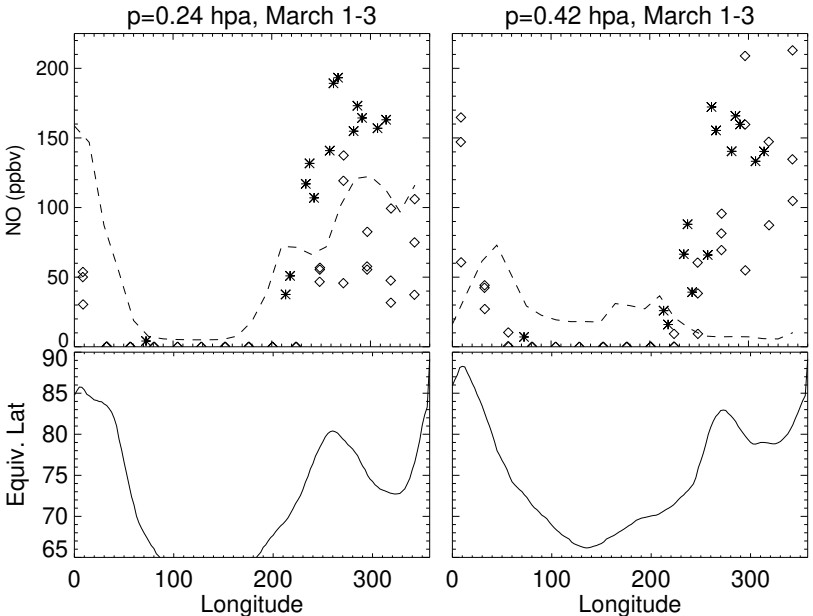

**Figure 10.** Comparison of WACCMX NO (dashed curve), ACE NO data (stars), and SOFIE NO data (diamonds) vs longitude. The WACCMX NO output is for March 2, the two datasets cover the period Mar 1-3. As in Figure 8, the geographic latitude is 81°N. Bottom panels show the calculated equivalent latitudes vs. longitude for this geographic latitude.

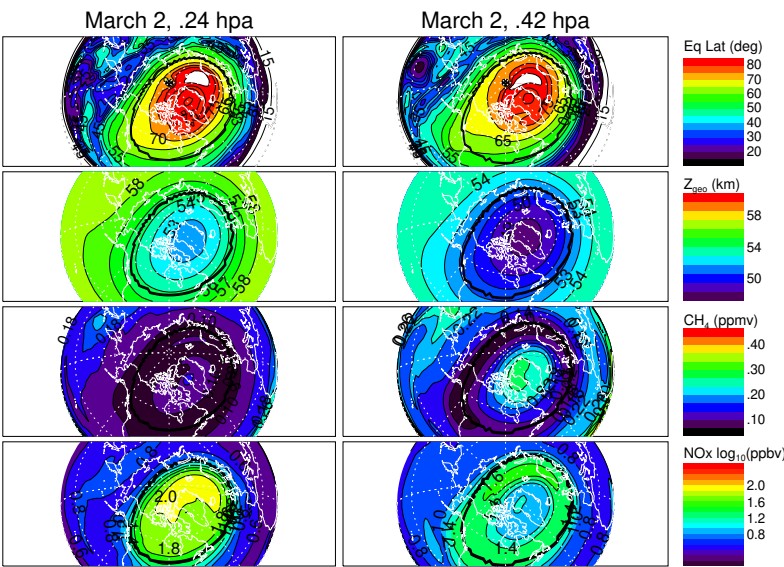

**Figure 11.** Polar plots of equivalent latitude (top row), geopotential height (Z, second row), CH$_4$ (third row) and NOx (bottom) from WACCMX. The contour interval is 5 degrees of latitude for equivalent latitude, 1 km for Z, and 0.2log$_{10}$(ppbv) for NOx. The orientations of the maps are such that the center of each map is 80°N, 90°W.

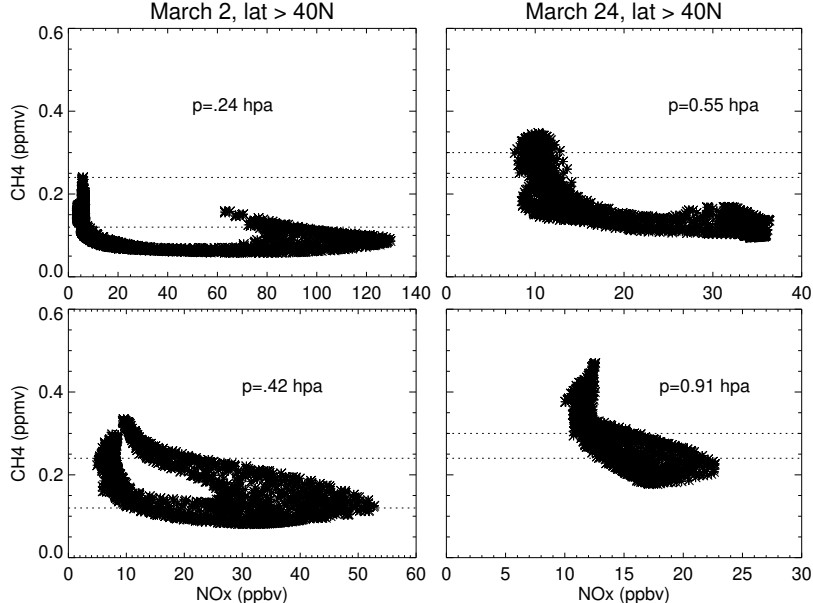

**Figure 12.** Scatter plot of WACCMX CH$_4$ vs. NO$_x$ for the indicated model dates and pressures. Note, the vertical range for CH$_4$ is the same for all 4 panels, but the horizontal ranges for NO$_x$ differ and are labeled separately for each panel. The horizontal dotted lines in each panel are CH$_4$ thresholds that are used to identify MLT NO$_x$; see text for discussion.

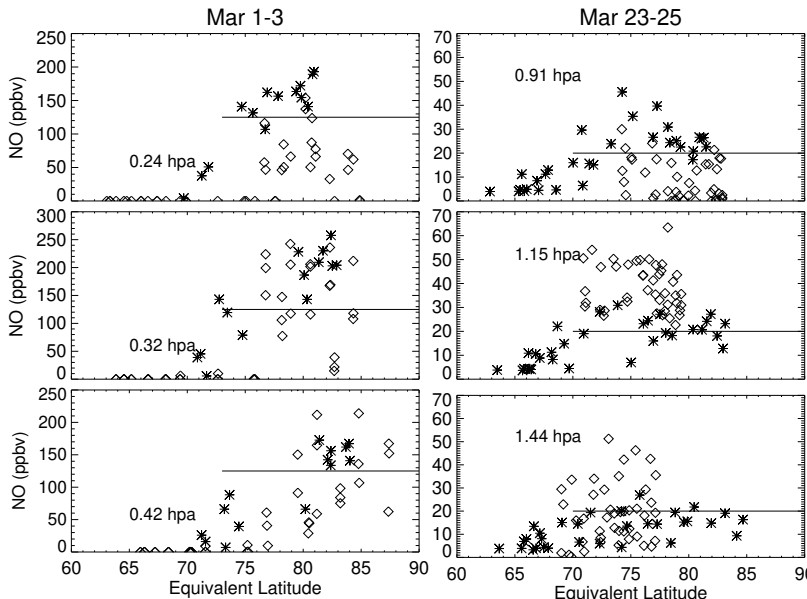

**Figure 13.** ACE (stars) and SOFIE (diamonds) NO data plotted vs equivalent latitude for the 3 pressures near the center of the layer as seen in Figure 8. The geographic latitude coverage is given in Figure 8. The horizontal lines in each panel are arbitrary constant values of NO used in a geometric estimate of the global amount of NO that might be consistent with the observations.

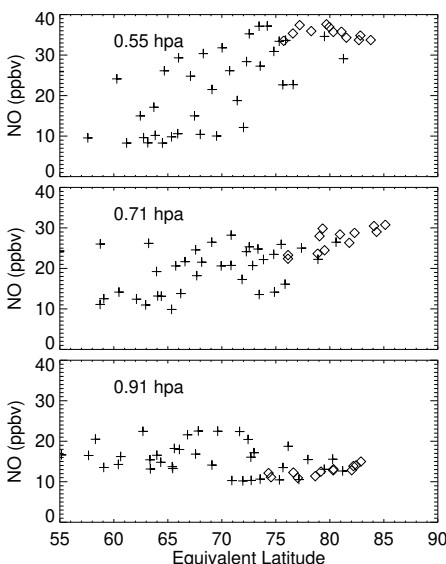

**Figure 14.** WACCMX NO, sampled according to the SOFIE (diamonds) and ACE (pluses) data shown in the right hand column in Figure 12. The pressures differ from those in Figure 12 due to the fact that the tongue of enhanced NO in WACCMX does not descend as low as is observed (see Figure 1).

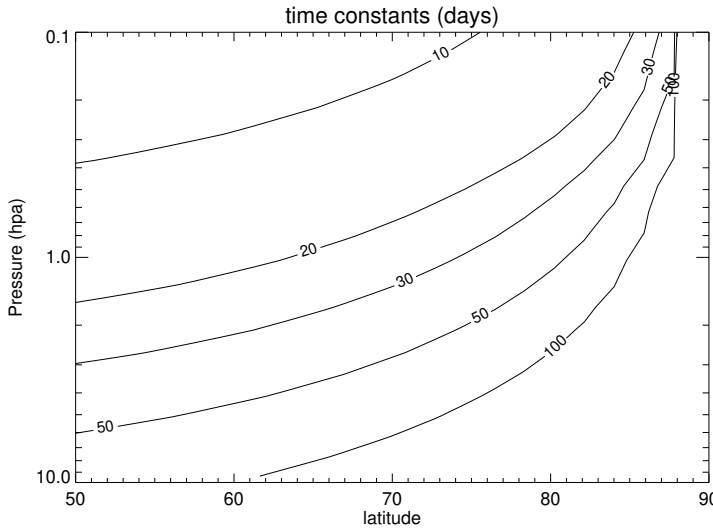

**Figure 15.** Diurnally averaged time constant against NO photolysis by UV sunlight as a function of latitude and pressure. Equinox conditions are assumed.

**Table 1.** Estimates of MLT NOx deposition in WACCMX and observations (in GM)

| Dates | vortex only | Equiv Lat 55-75 | Equiv Lat 60-75 | Equiv Lat 60-80 | high $CH_4$ threshold [1] | low $CH_4$ threshold [2] | geom. est.[3] |
|---|---|---|---|---|---|---|---|
| Mar 1-3 | .18 | .15 | .12 | .15 | .27 | .24 | .10 |
| Mar 23-25 | .11 | .17 | .12 | .15 | .27 | .18 | .082 |

[1] $CH_4$ threshold is .24 ppmv for March 1-3, and .3 ppmv for March 23-25

[2] $CH_4$ threshold is .12 ppmv for March 1-3, and .24 ppmv for March 23-25

[3] See text for discussion of uncertainites with this geometric estimate of the observed amount