# Peer review of "and 3-dimensional structure of the descent of mesospheric trace constituents after the 2013 SSW elevated stratopause event"

_Atmospheric Chemistry and Physics, 2021_

## Author Comment (AC1)

This paper investigates the downward propagation of NO after a sudden stratospheric warming / elevated stratopause event in early 2013 by analyzing results from WACCMX nudged into the MLT region compared to different observations and the NAVGM-HA data. The topic is of great relevance as sudden stratospheric warmings provide a highly variable and still not well understood source of NOx in the late-winter Northern hemisphere uppermost stratosphere and lower mesosphere. Unfortunately I found the paper rather unfocussed; it did not become entirely clear to me what the main scope and focus of the paper is. To evaluate the performance of a high-top model nudged by a high-altitude meteorological analysis compared to models nudged only up to the stratopause, in a particularly difficult dynamical situation? To analyze the dynamics of transport through the mesosphere after the SSW? To repudiate the idea that medium-energy electrons could be important for the energetic particle indirect effect? The second point appeared the most compelling and new to me as the analysis of longitudinal and latitudinal structures in NO and H2O after the warming as done in the paper provides new, highly relevant insights into downward propagation through the mesosphere in this dynamically disturbed situation. In particular it is shown that even in this model version nudged up to the mesopause, there are systematic differences in the downward transport through the mesosphere compared to the meteorological analysis which are related to (or expressed by) differences in the spatial distribution of areas of strong descent. Concerning the first point, it is shown that despite remaining differences, the WACCMX version nudged up to 90 km performs better than results from a previous model study nudging only up to the stratopause; this is hardly surprising, but important to point out. Concerning the third point, the evidence shown here is not entirely convincing to me. The authors argue that "in the absence of realistic meteorological forcing, one should be cautious about drawing firm conclusions about the role of medium-energy electrons", and I wholeheartedly agree with this statement; however, it can be turned around in the sense that "in the absence of a realistic representation of NO in the source region of the lower thermosphere, one should be cautious about drawing firm conclusions about the role of medium-energy electrons versus downward transport and mixing". In this sense, I recommend final publication after revisions mainly to make the focus and main conclusions of the paper clearer and more robust. Suggestions and more specific concerns are listed below, as well as a list of minor comments (typos and such).

We thank the reviewer for their careful review, which we believe has helped to improve the paper significantly. We have reworded the abstract and introduction to hopefully clarify things. We note here that the reviewer does correctly capture the 3 key points of the paper, and has them in order of priority. Regarding point #3, medium energy electrons- we are now trying to sidestep this since MEE's could encompass anything from 30 kev electrons which ionize at 90 km (about which we can't say) or higher energies which ionize as low as 60-70 km and for which there is no evidence in SOFIE. Our conclusions are guided by the recent publication by Duderstadt et al (JGR, 2021) who show no evidence for any NO response in the SOFIE data and comment that it may have to do with the latitudes of SOFIE's sampling. Our new figure 2 also clearly shows that both model and data have descent from 84 km to the mid-mesosphere and that it's only descent- no evidence for any jump or discontinuity in the data that might indicate production. We have removed all mention of the term "medium energy", but preserve our comments about lack of additional NO sources from ionization below 80-85 km since that most directly speaks to our observations and modeling.

As part of our overall response, there are a number of new references that we've added. For ease of reference, these are listed below; our specific responses follow this list.

1. Duderstadt, K. A., C.-L. Huang, Spence, H.E., Smith, S., Blake, J. B., Crew, A. B., Johnson,A. T., Klumpar, D. M., Marsh, D. R., Sample, J. G., Shumko, M., Vitt, F. M., Estimating the impacts of radiation belt electrons on atmospheric chemistry using FIREBIRD II and Van Allen Probes observations, J. Geophys. Res., https://doi.org/10.1029/2020JD033098, 2021.

2. Funke, B., Lopez-Puertas, M., Holt, L., Randall, C. E., Stiller, G. P. and von Clarmann, T., Hemispheric distributions and interannual variability of NOy produced by energetic particle precipitation in 2002-2012, J. Geophys. Res., 119, 13,565-13,582., doi:10.1002/2014JD022423.

3. Harvey, V.L., Randall, C. E., Hitchman, M. H. Breakdown of potential vorticity-based equivalent latitude as a vortex-centered coordinate in the polar winter mesosphere, J. Geophys. Res., 114, D22015, doi:10.1029/2009JD012681.

4. Hendrickx, K, Megner, L., Marsh, D. R., and Smith-Johnson, C., Production and transport mechanisms of NO in the polar upper mesosphere and lower thermosphere in observations and models, Atmos. Chem. Phys., 18, 9075-9089,  10.5149/acp-18-09750-2018, 2018

5. Perot, L., and Y. J. Orsolini, Impact of the major SSWs of February 2018 and January 2019 on the middle atmospheric nitric oxide abundance, J. Atm. Solar. Terr. Phys, 2021, in press.

6. Randall, C. E., Siskind, D. E., Bevilacqua, R. M., Stratospheric NOx enhancements in the southern hemisphere vortex in winter/spring of 2000, Geophys. Res. Lett., 28, 2385-2388, 2001.

7. Sinnhuber, M., Friedrich, F., Bender., S. The response of mesospheric NO to geomagnetic forcing in 2002-2012 as seen by SCIAMACHY, J. Geophys. Res., 121, 3603-3620, doi:10.1002/2015JA022284, 2016.

Our responses to specific points follow.

Abstract lines 8-9: this is only plausible if you assume that the sources of NO you included in your model for the MLT region are accurate. These presumably are photoionization and auroral electrons. However, from previous publications investigating MLT NO with the WACCM model, I would assume that this is not necessarily the case, as there appears to be evidence that the NO production by photoionization is too large (e.g., Siskind et al., 2019), while the parameterization for auroral electrons produces the NO peak at a rather high altitude (e.g., Smith-Jonsen et al., 2018). So it seems possible that the NO amount agrees reasonably well in a certain location and time due to a compensation of two antithetical error sources. Unfortunately the different NO formation mechanisms in the MLT – photoionization, auroral electrons, upper

boundary condition -- in the model version used here are not described in the paper, so it is not possible for the reader to consider this adequately.

Photoionization would not be relevant for winter high latitudes. Certainly the auroral electron source is. But there are other factors at play in the 90-100 km region, specifically eddy diffusion. Hendrickx et al (2018) did a complete study of this, which is now cited.

Abstract, line 10 – 13, "Despite the general success of WACCM in simulating mesospheric NO, …" this is a very positive way put it. A more critical assessment would be "Despite the general realistic temporal development of mesospheric NO in WACCM in the zonal averaged view, …" but this is also not quite true, as there appear to be significant differences in the downward motion in the lower mesosphere which are also observable in NO.

Reworded

Abstract, line 16: differences in the GW forcing are certainly to blame for a lot of problems in modelling middle atmosphere dynamics. However: what is your statement based on that the differences are "small"? Small compared to what? Maybe just leave out the "small". OK Also: is it possible that differences in wave-wave interaction between planetary-scale and gravity waves play a role here as well? The distribution of NO as shown in Figure 5 seems reminiscent of a planetary wave 1 forming between February 15 and March 1, both in WACCM and NAVGEM, though with a slightly different tilt. True, but that brings us back to GWs since the planetary wave distribution is prescribed by the analysis.

 Abstract, line 20: From the abstract, it is not quite clear what the aim of this study is – is it an investigation of the impact of zonal asymmetric behavior on downward motion after a sudden stratospheric warming? Or a quantification of the impact of SSWs on stratospheric or lower mesospheric NO? Or a demonstration that a model nudged into the MLT performs better? This never becomes clear in the paper either – maybe you could sharpen the focus a bit, both in the abstract and by discussing the aims and structure of the paper in a more concise way at the end of the introduction section (lines 20-26 of page 3).

Introduction was reworded at the end

Page 2, line 6: you could refer to the last WMO report (scientific assessment of ozone depletion, 2018) here – the impact of EPP on stratospheric NO and ozone is discussed there in the "polar ozone" chapter.

done

Page 2, line 15: There are quite a few references you could cite here using SCIAMACHY NO data to investigate the impact of electron precipitation on the mesospheric NO budget, the most concise are probably Bender et al., ACP, 2019 and Sinnhuber et al., JGR, 2016.

 Sinnhuber reference added

Pages 3-4, description of WACCMX: I am missing important information here to understand the performance of NO and transport in WACCMX specifically in the MLT region. For example, what is the vertical resolution in the upper mesosphere and around the mesopause? Is it already one-fourth of the scale height? From which pressure level on? By which mechanism is NO formed – is the same parameterization for auroral ionization used as described in Smith-Jonsen et al 2018? Is the upper boundary condition for NO used (probably not for WACCMX)? Is the same parameterization for EUV photoionization rates used as described in Siskind et al., 2019 (based on Solomon and Qian 2005), or has there been an update? When were the model runs started, and what were they initialized with? This last point is mentioned later on ("H2O was initialized by a December climatology") but not really clarified – is the model run started on December 1 of the winter? Or mid-December, or end of December? Are model results output on satellite footprints? Please clarify these points in Section 2.1.

Additional text added. Note, as part of this addition, we discuss the nudging and the time scales for the nudging. The idea is that the large scale meteorology in WACCMX is essentially slaved to NAVGEM-HA. The vortices are in the same place. Same planetary wave structure. This is borne out by the near identical temperature pattern in what is now Figure 7. What is different is the gravity wave forcing and this can (and does) affect the tracer distribution in WACCMX and can cause them to deviate from observations.

Page 5, line 5: is the model also sampled at the satellite overpasses, or is this strictly a zonal average at the latitudes of the observations? How are model data output – as zonal averages, or as fields of the full model grid at some specific time(s) of day? Please clarify (possibly in Sec 2.1)

Additional text added (yes, sampled at the SOFIE longitudes and then daily averaged)

Page 5, line 7: "are transported downward …" in the absence of chemical loss, that is. Considering the lifetime of NO in the high-latitude winter this is probably a justified zero-order assumption (and you do discuss this point later on), but you should make a statement about it here.

Text now says "descending"

Page 5, line 8: Looks like 0.2-0.3 hPa in SOFIE, 0.1-0.2 hPa in WACCM to me. However, it is quite difficult to read this from the contour figures. Why not provide profile plots for January 1 and February 15? Sorry, we didn't save the H2O from January and the co-author who did that work has since left NRL. We added a comment to that effect in the text.

Page 5, line 8: please provide date and doy for each period you discuss here; that would make the comparison with the figure much easier to follow. "Middle of February" presumably is around doy 45?

done

Page 5, line 10: "could differ" erase the "could" – they do differ significantly.

Section was extensively rewritten

Page 5, lines 14-15, "… it is unclear to what extent these higher altitude differences are relevant to the present study". Well as you do not include MEE, your NO presumably is formed by auroral electrons and EUV photoionization in these higher altitudes, and transported or mixed down to 0.01 hPa. That values there agree reasonably well with observations could argue a compensation of the too-low thermospheric values by more efficient mixing in the lower thermosphere. Your argument here appears to be that you only investigate the transport from 0.01 hPa into the upper stratosphere / lower mesosphere, and this is therefore not relevant. That is a fair point, but you should anyway discuss this point in a bit more detail here, and make more clear that this does not mean that thermospheric NO production in WACCMX is generally well reproduced. One feature I am missing in the discussion here is the apparent MLT upwelling in early January around the SSW. This seems to be strongest around day 8-10 in SOFIE, around days 10-15 in WACCMX, and the strengths of the upwelling appears to be different as well. In SOFIE, the 200 ppb isoline moves up from 0.01 hPa on January 1 to 0.004 hPa on January 10; in WACCMX, the 200 ppb isoline moves up from 0.01 hPa on January 1 to 0.002 hPa on (probably) January 12. The 200 ppb isoline in WACCMX thus covers a larger vertical area in a shorter period of time to reach 0.1-0.2 hPa around day 40. So downward transport after the event throughout the upper and mid-mesosphere appears to be faster in WACCMX.

The section was extensively rewritten. The new figure 2 shows the upwelling and the new text discusses this. It also mentions the somewhat more rapid descent in WACCMX as noted by the reviewer. At the same time, there really is no evidence for any contribution of air above about 0.004 hPa to the middle mesosphere and below. As we note in the text, the descending tongue of NO is peeled off the bottom of the layer that initially sits above 80-85 km. This is consistent with work we did many years ago (Randall et al., 2001) and that citation is added. As a consequence of all this, and as noted in our response to Reviewer 2, we have therefore changed the description of the NO in Section 5 from MLT NO to "upper mesospheric" since there is no direct evidence of thermospheric NO in the feature that we are modeling in this paper. Ultimately, the reservoir of enhanced NO that sits above 80-85 km may well have its origins in the thermosphere, but that is beyond the scope of this paper.

Page 5, lines 18-25 "Overall, the good agreement between calculated and observed NO at 0.1-0.2 hPa … Our results therefore suggest that for this specific period …, an additional odd nitrogen source … is not required. " As I pointed out above, if you look closely at NO in the source regions of the lower thermosphere, and at the temporal evolution of NO before it reaches the lower mesosphere, there are quite a few differences to the observations; too many to draw firm conclusions about the different sources of NO I think, even for early 2013. Of course you can speculate about this, but the evidence does not appear compelling.

Again, new Figure 2 hopefully addresses this.

Page 6, lines 3 to 16, Figure 2: I'm not quite sure what the purpose of this comparison is in respect to the aims of the paper. If the main purpose is to compare against the Orsolini et al results, you should include their data in your figure. However, if the purpose is to show that WACCMX performs better than WACCM as used by Orsolini et al, you should include

observations as a benchmark as well. This is now figure 4 and we did add some data from Orsolini et al's figure to flesh this out a bit as suggested.

Page 6, line 27 "remains at lower pressure" of about 0.2 hPa. Clarified: "closer to 0.2 hpa"

Page 6, lines 27-28: "…have descended another scale height …" it is not quite clear what the reference here is – March 1 or February 15? The feature is now at 0.6 hPa in NAVGEM  respectively 0.7 hPa SOFIE, at 0.4 hPa in WACCMX. This is less than a scale height (one order of magnitude in pressure) even in SOFIE, certainly much less in WACCMX. Please clarify. Clarified.

Section 4: This is really an interesting analysis. Thank you!

Page 7, line 34, discussion of Figure 5: just a note – what I see on March 1, very clearly in NAVGEM, less pronounced in WACCM, is a planetary wave-1 structure with a zone of large-scale descent presumably related to the polar vortex displaced to Greenland.

Planetary wave structures in tracers (but not in dynamical fields) will differ between NAVGEM and WACCM due to unresolved forcing from gravity waves that differs. On March 1[st], WACCM descent is not over Greenland. The green color (low H2O) forms a ring around Greenland. Greenland shows wetter air. NAVGEM however, has descent over Greenland. NAVGEM and WACCM are very different- this difference is the focus of the last paragraph on page 8.

Page 8, line 26: for better readability, and to ensure the reader can follow you without having to read lots of other papers, you could provide equation 4 of Siskind et al 2010 and equation 3.5.2c of Andrews et al 1987 here as well. Text from Siskind et al 2010 inserted.

Page 9, lines 15-16, "A truly comprehensive examination of the causes of these differences is beyond the scope of the present study"; I accept that it is unlikely that you will clear this question within this study, but it seems to me that you could go one step further in evaluating your statement that the reason for the differences are more likely due to the representation of GWs than due to planetary-scale waves. E.g., you could test whether the planetary-scale waves really are represented consistently in WACCMX and NAVGEM. Figure 6 (lower right panel) seems to confirm this assumption, but this is only one latitude; e.g., you could easily calculate amplitudes and phases of planetary waves 1, 2, … for this date for a wider latitude range.

 It's not an assumption- by design from the nudging on 1 hour time scale, the large scale dynamical features in WACCMX are essentially slaved to NAVGEM-HA.  This is now mentioned in Section 2.1. Figure 7 is completely consistent with that.

Page 10, line 7 "shows good agreement" …I wanted to suggest to compare mean/median peak values and peak altitudes, however considering the strong longitudinal variation as shown in Figure 9 this is probably not very meaningful. Maybe you could clarify somehow that it is a qualitative agreement of peak values and altitudes in those profiles which show enhanced values.

 Clarified as suggested

Page 10, lines 13-14: actually the peak of SOFIE and ACE as shown in Figure 8 seems to lie around 0.3 hPa (ACE) respectively between 0.3 hPa and 0.4 hPa (SOFIE), definitely not below 0.4 hPa.

WACCMX has grid points at 0.32 and 0.42 hPa. We think 0.42 hPa is closest to the peak and added a statement in the text to this effect (top of page 11).

Page 10, lines 26-27: As the relationship between equivalent latitudes and enhanced NO / low H2O emphasizes downwelling of upper mesosphere air in the polar vortex, it is quite puzzling that this appears to be better reproduced at the higher altitudes than at the lower altitudes – I would assume the vortex to be better represented lower down. Question – equivalent latitudes here have been calculated from NAVGEM data? Could you calculate those from WACCMX? That is, is it possible that the vortex itself is shifted in WACCMX compared to NAVGEM? Or is it larger, or is the edge more diffuse?

The equivalent latitudes are calculated from WACCMX, and this is now stated in the paper. The difference between enhanced NO and equivalent latitudes must be due to transport from gravity waves which are not accounted for in the nudging.

Page 12, Lines 28 and following, derivation of "geometric estimate" based on SOFIE and ACE: the procedure here is not entirely clear from your description. As I understood it, you just draw a line by hand which provides an approximate average of ACE and SOFIE NO values and equivalent latitude coverage extrapolated to 90° in this pressure level (so assuming homogeneous descent in the vortex), and then integrate this line. This appears to be rather imprecise considering the large differences, e.g., between SOFIE and ACE, and your description of non-isotropic descent above. As a first-order approximation it might be justified to do that. However, you could make more clear the limitations of this approach, and derive an error range based on the observed variability of NO as well as by excluding the high-latitude area not covered by data.

We added some text which hopefully clarifies how we obtain an error estimate- either from misjudging the thickness of the layer, or its equatorward extent (factor of 2 as an upper bound, as noted at the bottom of the paragraph).

Page 12, lines 34-35: … do not spread to such low equivalent latitudes as suggested by the model … I would say that the relationship between equivalent latitudes and enhanced NO values in the model is not as clear as in the observations; which again raises the question whether the vortex in WACCMX was formed somehow differently than in reality, see my comment above (Page 10, lines 26-27). No, it has to be gravity waves which cause transport to deviate from expectations based purely on planetary waves and the vortex.

Page 13, lines 22-23: "However, the global totals are quite close. Certainly the immediate conclusion one draws is …." I don't really follow your conclusion here; if you want to make a bold statement like this, you would have to a) derive a robust error range of the observational (geometric) estimate, and b) show that NO agrees well in the source regions of the lower thermosphere and in the temporal behavior from before the event onset to the lower mesosphere.

Considering b), there are problems with both cases as discussed above in my comments to page 5. Your second sentence "Certainly the immediate conclusion is that it is hard to argue …." is also formulated in a rather indirect way; that could be put clearer. Why not just state "We conclude that WACCMX/NAVGEM-HA represents the NOx descent to the lower mesosphere reasonably well after this event, and no additional source of NOx is needed to reproduce the total amount of EPP NOx during this time." Reworded as suggested.

Page 13, line 26: see my comment to gravity waves versus planetary wave forcing (page 9, lines 15-16) See responses above

Page 14, line 7-10: See my comments to page 5 and to page 13, lines 22-23; I don't really follow this conclusion; I think if you want to draw a robust conclusion on the MEE versus transport issue, you have to show that NO in the source regions in the lower thermosphere is well reproduced. I agree with your last sentence, that "in the absence of realistic meteorological forcing, one should be cautious about drawing firm conclusions about the role of medium-energy electrons "; but this sentence can be turned round as well to "in the absence of a realistic representation of NO in the source region of the lower thermosphere, one should be cautious about drawing firm conclusions about the role of medium-energy electrons versus downward transport and mixing".

We removed the last sentence "in absence of…. Etc etc".

**Minor comments regarding typos and such**

Page 2, line 22: Full stop missing. Corrected.

Page 2, line 23: the best comparison for this is provided in my opinion in the full MIPAS NOy timeseries as shown in Funke et al., 2014 yes, thank you- added both here and at the end of the paragraph where we discuss our estimates from WACCM.

Page 3, line 16: double full stop. Corrected.

Page 4, line 31: "A similar" not "As similar" Corrected.

Page 5, line 4: please format "1E-5". fixed

Page 5, line 11: "proior"? Corrected.

Page 5, line 28: "much less" better "much smaller"? section has been significantly rewritten

Page 5, line 32: "much less" better "much weaker"? ibid

Page 10, line 15: "too low" Corrected.

Page 10, line 15: please format "1e-14" fixed

Page 12, line 10 "… that could to used …." Should be "… that could be used …." fixed

Page 12, line 33: "… can be compared the WACCMX data …." Should be "… can be compared to the WACCMX data …" corrected

Page 13, lines 19-20 just strike out "as discussed above, the difference is immaterial" OK

Figure 11: Note x-axis labels of the lower panels are overlapping into the lower panels. Also the panels on the right-hand side seem to be shifted vertically compared to the left-hand panels. Please correct. corrected

Table 1: Please clarify in the caption of Table 1 that "observed" refers to the last column "geom. est.", and that this is an estimate which carries a large uncertainty.

Changed header in Table and added a footnote at the bottom. As noted above, we added some hopefully clarifying text

:wq

---

## Author Comment (AC2)

This paper examines the descent of nitric oxide (NO) and water vapour in the northern high latitudes during the stratospheric warming (SSW) of 2013, using a version of WACCM-X nudged to the high-altitude NAVGEM analyses, extending to the mesopause. Results are compared to older simulations with WACCM driven by the MERRA reanalyses extending only to the stratopause region.

The paper shows that when constrained by reanalyses extending higher, the model better reproduces the evolution of NO seen in satellite observations, if not in all the details at least in the mean total transport into the stratosphere.

The paper is relatively clear and well-written, although it is a bit long. It is worthy of publication in ACP after a couple of major comments and minor comments are properly addressed.

We thank the reviewer for their comments. The first part of the paper has been substantively rewritten as a response. A number of new citations have been added as we discuss below. Also, and importantly, a new figure (Figure 2) is added to track more clearly how the NO from the upper mesosphere ends up in the lower mesosphere. Based upon this we conclude that the tongue of descending NO is peeled off the bottomside of the NO layer that sits above 80-85 km and that "upper mesospheric" rather than "MLT" more properly describes that. This is consistent with similar conclusions reached by Randall et al., (2001) and that citation is now added. More specifics on this issue are offered below.

Note, as part of our overall response, there are a number of new references that we have added. For ease of reference, these are listed below; our specific responses follow this list.

1. Duderstadt, K. A., C.-L. Huang, Spence, H.E., Smith, S., Blake, J. B., Crew, A. B., Johnson,A. T., Klumpar, D. M., Marsh, D. R., Sample, J. G., Shumko, M., Vitt, F. M., Estimating the impacts of radiation belt electrons on atmospheric chemistry using FIREBIRD II and Van Allen Probes observations, J. Geophys. Res., https://doi.org/10.1029/2020JD033098, 2021.

2. Funke, B., Lopez-Puertas, M., Holt, L., Randall, C. E., Stiller, G. P. and von Clarmann, T., Hemispheric distributions and interannual variability of NO$_y$ produced by energetic particle precipitation in 2002-2012, J. Geophys. Res., 119, 13,565-13,582., doi:10.1002/2014JD022423.

3. Harvey, V.L., Randall, C. E., Hitchman, M. H. Breakdown of potential vorticity-based equivalent latitude as a vortex-centered coordinate in the polar winter mesosphere, J. Geophys. Res., 114, D22015, doi:10.1029/2009JD012681.

4. Hendrickx, K, Megner, L., Marsh, D. R., and Smith-Johnson, C., Production and transport mechanisms of NO in the polar upper mesosphere and lower thermosphere in observations and models, Atmos. Chem. Phys., 18, 9075-9089, 10.5149/acp-18-09750-2018, 2018

5.  Perot, L., and Y. J. Orsolini, Impact of the major SSWs of February 2018 and January 2019 on the middle atmospheric nitric oxide abundance, J. Atm. Solar. Terr. Phys, 2021, in press.

6.  Randall, C. E., Siskind, D. E., Bevilacqua, R. M., Stratospheric NOx enhancementsin the southern hemisphere vortex in winter/spring of 2000, Geophys. Res. Lett., 28, 2385-2388, 2001.

7.  Sinnhuber, M., Friedrich, F., Bender., S. The response of mesospheric NO to geomagnetic forcing in 2002-2012 as seen by SCIAMACHY, J. Geophys. Res., 121, 3603-3620, doi:10.1002/2015JA022284, 2016.

Response to Major comments

1) I wonder the role of downward transport from the NO main reservoir into the mesosphere near 90-100 km that is glimpsed from the observations and whether the model captures it. Orsolini et al. (2017) or Limpasuvan et al (2016) showed a short-duration downward transport, diagnosed in the TEM formalism, driven by transient planetary wave activity following the onset.  That westward planetary wave forcing driving the downward motion was able to overwhelm the eastward gravity wave drag.  That 2017-paper used WACCM nudged to MERRA and I wonder if there is a similar w* signature in the simulations discussed in this paper.  The implication is that this descent might still be underestimated in the model simulations, in a region where the constraint from NAVGEM is relaxed.

It would hence be of interest to see w* higher than 0.01 hPa in Fig 7 (like in Fig 1), from the time of onset and onwards. The authors argue that what happens at these higher altitudes does not influence the descent lower down, based on examination of Fig 1. However, it seems that SOFIE indicates higher values of NO than WACCM-X at 0.001 hPa and high mixing ratios do seem to migrate downwards (Fig 1, in second part of January).  I understand that there might be a bias due to initialisation, but it might be possible to rescale values to track the rate of descent. It may also be that the descent is confined in longitudinal sectors, constrained by the presence of the planetary waves, and not entirely captured by a zonal-mean like Fig 1, as the authors have also diagnosed during other events in other publications.  In other words, can it be confirmed that a strong transient downward transport near 90-100km is not occurring during this 2013 event shortly after the onset in the simulation or, if it occurs, that it does not impact NO much lower down?

As we now discuss in the context of the new Figure 2, the tongue of the descending NO that is the focus of the paper does not come from 0.001 hPa. We do acknowledge in the text (page 6) that there are differences between WACCMX and SOFIE at the higher altitudes, but it is our assessment that those are not relevant to the present study. We did look at specific longitudes to confirm this (not shown). Certainly the large reservoir of NO that persistently resides in the 85-100 km region could well be considered "lower thermospheric"; however, the details are beyond the scope of the present study. We did add a citation to Hendrickx et al. (2018) who looked into some of this. As a consequence, subsequent to the discussion of Figure 2, we replaced all

mention of "lower thermosphere" or "MLT" in favor of "upper mesosphere" to more rigorously describe the NO we are studying (cf. throughout Section 5).

2) Some more details about how the equivalent latitude is calculated would be useful and would fit in Section 2. Is the PV from WACCM-X used or from NAVGEM, and do you use spatial filtering first to remove small-scale PV anomalies (linked to GW drags) which are common at these altitudes? Discussion added in Section 5 with citation added to Harvey et al (2009).

3) Some climatological validation of WACCM-X-NAVGEM NO climatology against SOFIE at the top level shown in Fig 1, near 0.001 hPa, based on several years of available simulations and observations, would be highly valuable to support with confidence that the WACCM-X distribution of NO is realistic at the top of the mesosphere. Given the dependence upon geomagnetic activity and EPP, I realize that enough simulated years might not be available.

As noted above, we have concluded that those differences have no immediate consequence or relevance for the tongue of descending NO that is the subject of the paper.

Minor comments

P2, line 17: You could add Odin/SMR to the list of satellite instruments, since data from SMR is referred to later in the paper, when describing the descent of NO during SSWs. In fact, Odin/SMR has also been used to describe the descent of dry air through the mesosphere during SSWs. done

P2, line 31: In fact, the 2018 event was not an elevated stratopause event, but the 2019 event was. Corrected- we removed the Wang et al., reference and added Perot and Orsolini's recent paper

P3, line 33: Some details about the initialisation would be helpful: how is the model initialised at levels above where the NAVGEM data is used; initialisation of chemical species (in particular NO and $H_2O$ but also other species) could be addressed in this section (it is referred to only later in the paper). Text added in Section 2.1

P7, line 12 : the sentence is a bit confusing since the $H_2O$ meridional gradient is negative, with drier air at the pole. Sorry, no- in WACCM the lowest value of $H_2O$ is sub-polar. From 70 to the pole, the air gets wetter- a positive meridional gradient.

P9, line 7: the sentence is a bit unclear: "the 0.1-0.2 hpa equatorward flow is seen in NAVGEM-HA moving downward…". Hopefully clarified "at 0.1-0.2 hpa, equatorward flow is seen…"

P9, line 26: a "zero wind line" might not be the most appropriate term. What is meant is a line where w* goes to zero. This expression is mostly used in connection with zonal wind, and w* is not an actual wind. "encounters a level where w*goes to zero…" or something like that (?)

Yes, corrected as suggested (both at the beginning of section 5 and a couple of places at the end of Section 4)- it now reads "zero isoline" or "zero line in the descent" or "where w* goes to zero"

P10, line 12: measure at the same latitude or sample the same latitude (not are at the same latitude). Corrected: "sampling"

P13, line 30: A word about the potential causes of the stronger equatorward dispersion of NO in WACCM-X would be useful. We added reference to Figure 8. Also "spreads" was not the best word- is "advected" since that is what the TEM circulation characterizes.

Figure 4 is missing a color scale. Added

Figure 7: could a streamfunction help visualization of the different circulation cells ? Perhaps; hopefully some of our clarifications make it easier to follow.

Typos/English

P4, line 4 : Sassi (2020) Woops- sorry, there were supposed to be two Sassi references, one to a 2018 paper and one to a 2021 paper. Hopefully now clarified/corrected.

P4, line 21: short term corrected

P4, line 31: A similar figure Corrected

P5, line 8 : check consistency about use of hPa and hpa (also other places in the paper) OK

P7, line 2: use daily rather than diurnally to be consistent with captions Done.

P10, line 15: too low Corrected

P13, line 20: is not present (?) "does not present itself". Corrected.

---

## Author Response (AR2)

Response to reviewers (reviewer comments in black, authors in red)

To Reviewer #1

We made the technical corrections and removed the phrase "continue to" in our conclusions concerning the need, or lack thereof, of in-situ production from energetic electrons. I should note, that Siskind et al (2015) reached the same conclusion for the 2009 SSW event and Shepherd et al (2014) reached the same conclusion for the 2006 SSW event. It does seem like the beginning of a pattern. However, the third sentence of the paragraph discusses alternative scenarios where energetic electron precipitation would be important and why the present simulations may not be of complete general applicability.

To Reviewer #2

Motivated by the reviewer's comments, we looked further at the question of where the transport originates. We have added new text on page 6 concerning the possible occurrence of longitudinally-isolated occurrences of transport from above 90 km into the lower mesosphere and document that we did not find any. I do believe this discussion strengthens our argument. Based upon that, we continue to be very reluctant to add another figure on w* as we the evidence continues to support our argument that transport from above 90 km is not relevant to this event. And even if descent from above 90 km had been occurring in specific longitude sectors, w* would not be an appropriate diagnostic is it is, by definition, a zonal mean variable.

Responses to other comments (reviewer comments in black, authors in red)

P4, line 15: In fact, the SD-WACCM simulations of the same 2012/2013 event in Orsolini et al. (2017, cited) were also made with both Pr=2 and Pr=4. Their conclusion was that the high diffusion run did enhance the transport of NO in the mesosphere, where GWs break, but not in the LT.
We added mention of Orsolini's work. While they may differ in the specific details of where the transport was perturbed, both their work and Hendrickx are consistent in arguing against our use of higher diffusion in our simulations for this event.

P14, line 20): the authors argue that the equatorward transport is stronger in WACCM-X than what is inferred from SOFIE or ACE. It is mentioned "as discussed in the context of Figure 8…" Shouldn't Figs 13 and 14 be rather mentioned here? Or do the authors refer specifically to the WACCM-X equatorward meridional velocities in Fig 8 between 1.0 and 0.1 hPa (incidentally, similar to NAVGEM) ? Please clarify.
Yes, good point, thank you. We've added mention of those two figures.

Typos/English/Wording
P2, line 7: (Langematz and Tully, 2018) corrected
P2, line 24 (also P3, line 6): I wonder if the term "long range transport" is the best choice of terminology since it has a connotation of long-range (e.g., pollutant or aerosol) horizontal transport, esp. for readers of ACP. Here it is meant deep vertical transport across atmospheric layers. We added the word "vertical" to clarify.
P5, line 15: remove "bf" before Figure 1 fixed

P17, line 12: Smith-Johnsen corrected, sorry for the misspelling
Caption of Fig 4: Indicate that stars and diamonds taken from Orsolini et al. (2017) are from SMR satellite observations. done
Caption of Fig 2: remove "This tracks" done